# Switches, stability and reversals in the evolutionary history of sexual systems in fish

Susanna Pla[1,4], Chiara Benvenuto [2,4], Isabella Capellini [3✉] & Francesc Piferrer [1✉]

Sexual systems are highly diverse and have profound consequences for population dynamics and resilience. Yet, little is known about how they evolved. Using phylogenetic Bayesian modelling and a sample of 4614 species, we show that gonochorism is the likely ancestral condition in teleost fish. While all hermaphroditic forms revert quickly to gonochorism, protogyny and simultaneous hermaphroditism are evolutionarily more stable than protandry. In line with theoretical expectations, simultaneous hermaphroditism does not evolve directly from gonochorism but can evolve slowly from sequential hermaphroditism, particularly protandry. We find support for the predictions from life history theory that protogynous, but not protandrous, species live longer than gonochoristic species and invest the least in male gonad mass. The distribution of teleosts' sexual systems on the tree of life does not seem to reflect just adaptive predictions, suggesting that adaptations alone may not fully explain why some sexual forms evolve in some taxa but not others (Williams' paradox). We propose that future studies should incorporate mating systems, spawning behaviours, and the diversity of sex determining mechanisms. Some of the latter might constrain the evolution of hermaphroditism, while the non-duality of the embryological origin of teleost gonads might explain why protogyny predominates over protandry in teleosts.

[1] Institut de Ciències del Mar, Spanish National Research Council (CSIC), Barcelona, Spain. [2] School of Science, Engineering and Environment, University of Salford, Salford, UK. [3] School of Biological Sciences, Queen's University, Belfast, UK. [4]These authors contributed equally: Susanna Pla, Chiara Benvenuto. ✉email: I.Capellini@qub.ac.uk; piferrer@icm.csic.es

Sexual reproduction is a unifying feature of eukaryotes[1] and yet it is extremely diverse[2]. Sexual systems (also known as "sexual patterns"), defined as the pattern of distribution of the male and female function among the individuals of a given species, vary from separate fixed sexes (known as gonochorism in animals and dioecy in plants) to simultaneous hermaphroditism (each individual produces both male and female gametes at the same time). These two sexual systems can be viewed as the extremes in a sexually plastic gradient[3] of intermediate systems (sequential hermaphroditism) and mixed systems (coexistence of males and/or females with hermaphrodites)[4,5]. Sexual systems have a profound influence on individuals' mating success and fitness[6], population sex ratios and effective sizes[7], as well as colonisation events and habitat use[8]. As a result, sexual systems influence the population dynamics and resilience to natural and anthropogenic stressors of ecologically and commercially important species that are often endangered or overexploited[9].

Hermaphroditism is predominant in flowering plants (angiosperms)[10], where 94% of the species have male and female sex organs in the same individual/flower, and it is widespread in invertebrates and teleost fish (the only vertebrates to exhibit hermaphroditism[11]), totalling 5% of animal species or up to ~30% if insects are excluded[12]. While this diversity suggests multiple evolutionary transitions between sexual systems in response to selection, current evolutionary models on the adaptive advantage of different sexual systems explain little about how and why sexual systems evolve and thus their large-scale distribution across the tree of life. This might indicate that adaptive predictions alone fail to fully explain why some sexual forms evolve in some taxa but not others (Williams' paradox)[4,13]. Therefore, unravelling the evolutionary history of sexual systems and quantifying how frequently and in what direction transitions occur is key to revealing which sexual systems are evolutionarily labile or stable, elucidating how one changes into another over evolutionary time, and identifying the environmental, genetic and developmental drivers favouring or opposing these changes. Yet, our understanding of how sexual systems evolve is still limited, particularly in animals.

Theoretical models, initially developed for plants, suggest that simultaneous hermaphroditism and dioecy are evolutionary stable conditions that are retained over a long evolutionary time and unlikely lost once evolved, while mixed sexual systems represent evolutionary intermediate stages[4,5,14] (Fig. 1).

Simultaneous hermaphroditism is likely the ancestral state in angiosperms from which dioecy, a rare sexual system in plants[6], has evolved independently several times, possibly to avoid inbreeding[15,16]. Theoretical models predict that separate sexes in plants evolve from hermaphroditism in different ways: (1) primarily through the intermediate state of gynodioecy[17], a common sexual system in plants that occurs when a male-sterile mutant invades a hermaphroditic population resulting in the coexistence of hermaphrodites and females; (2) through androdioecy, a less common system[18,19] in which mutations resulting in female sterility lead to the coexistence of hermaphrodites and males; (3) via trioecy, i.e., the coexistence of hermaphrodites, males and females, which is very rare; and (4) less frequently, via a direct transition[6,10] (Fig. 1). However, in animals, no evidence of a direct transition between hermaphroditism and gonochorism exists. Once gained, dioecy was believed to be an irreversible condition[20], a conclusion based on the assumption that returning to a simultaneous expression of male- and female-specific genes would likely produce contrasting effects on sex-specific physiology. Recent studies, however, reject this claim in plants, as phylogenetic reconstructions of direct transitions from dioecy/gonochorism to simultaneous hermaphroditism have been documented[10,21].

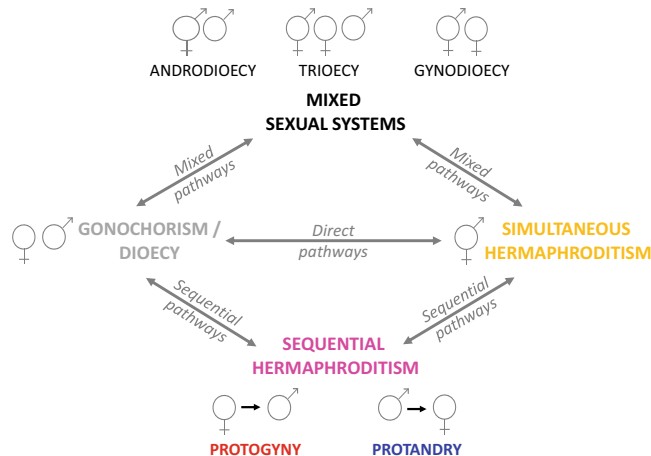

**Fig. 1 Theoretical framework for the evolution of sexual systems.** Illustration of potential evolutionary transitions between gonochorism (in grey) and simultaneous hermaphroditism (in yellow) via mixed systems (mixed pathways) as described in plants and some animals; via sequential hermaphroditism (sequential pathways: protogyny in red and protandry in blue) as recently suggested[4]; or without intermediate states (direct pathways) as proposed for plants[6]. Double-headed arrows indicate theoretical pathways.

The same theoretical framework with mixed pathways has been proposed also for animals where, in contrast to plants, gonochorism is the most common sexual system, androdioecy is more common than gynodioecy[5,14] and trioecy is very rare[22]. However, several reproductive characteristics in plants differ substantially from those in animals[23], albeit similarities can be found in some invertebrates[24]; hence, different theoretical frameworks are required (Fig. 1). Furthermore, evolutionary transitions between sexual systems in teleost fish (~34000 species, comprising the overwhelming majority of the ray-finned fishes, Actinopterygii)[25], might be less likely to occur via a mixed pathway (Fig. 1) given that in this group only a few killifish species of the genus *Kryptolebias* (formerly *Rivulus*) are truly androdioecious[5,26,27]. Beyond teleosts, the presence of gynodioecy and trioecy among vertebrates is still debated in the jawless hagfish *Myxine glutinosa* (Myxini)[14,28]. Recently, sequential hermaphroditism has been suggested as a possible intermediary state that may facilitate evolutionary changes between gonochorism and simultaneous hermaphroditism[4] (Fig. 1). However, phylogenetic studies on the evolution of hermaphroditism at large scale do not typically discriminate between the different forms of hermaphroditism and treat the sexual systems as a binary trait[29]. Thus, we currently have no robust large-scale study on the evolution of sexual systems in animals and we do not know whether sequential hermaphroditism represents an evolutionary intermediate stage between gonochorism and simultaneous hermaphroditism, whether protogyny and protandry act equally as transitional forms between the two, and whether gonochorism and simultaneous hermaphroditism are evolutionary stable conditions in animals as they are in plants.

The evolution of hermaphroditism in animals has mostly been interpreted in the context of its adaptive advantages relative to gonochorism, as proposed by the low density and the size advantage models[30]. The former predicts that simultaneous hermaphroditism evolves under low population densities and/or low dispersal capacity as, in these conditions, individuals with this sexual system can maximise their chances of securing a mate compared to sex-changing or gonochoristic individuals[31]. Note, however, that the advantages of self-fertilizing simultaneous hermaphrodites may be offset by an increased risk of inbreeding. The size advantage model proposes adaptive explanations for the

**Table 1 Predictions of associations between most common sexual systems (pattern of distribution of the male and female function among the individuals of a given species), mating systems (pattern of sexual interactions that take place considering the number of males and females involved in reproduction), adult sex ratio, size of mates, and spawning behaviour (how the two sexes interact to release the gametes) in teleosts.**

| Sexual System | Mating system | Adult sex ratio | Size of mates | Spawning behaviour |
|---|---|---|---|---|
| GONOCHORISM (G) Individuals reproduce as one sex throughout their lifetime (male or female) | Monogamy (pair bond) or random pairing Promiscuity Harem polygyny or temporary lek-like systems* | Variable | Males smaller, similar or larger than females | Pair spawning (pair of individuals) Group spawning** |
| PROTOGYNY (PG) Female-first sequential hermaphroditism: individuals first reproduce as females, change sex once with increasing size/age and then reproduce as males | Harem polygyny or temporary lek-like systems* Promiscuity | Female-biased | Males larger than females | Pair spawning (pair of individuals) Group spawning** |
| PROTANDRY (PA) Male-first sequential hermaphroditism: individuals first reproduce as males, change sex once with increasing size/age and then reproduce as females | Monogamy (pair bond) or random pairing | Male-biased | Females larger than males | Pair spawning (pair of individuals) Group spawning** |
| BIDIRECTIONAL (BD) Individuals can change sex more than once, in either direction, throughout their lifespan, usually starting from PG | Monogamy (pair bond) or random pairing Harem polygyny or temporary lek-like systems* | Female-biased | Males larger than females | Pair spawning (pair of individuals) |
| SIMULTANEOUS (SH) Individuals produce gametes of both sexes at the same time or in a short period of time | Monogamy (pair bond) or random pairing | 1:1 | Males similar to females | Pair spawning (pair of individuals) |

*multiple females in a territory defended by a male.
**multiple males and multiple females or one female with multiple males.
This general set of predictions is applicable to most species, but exceptions are found in species with less common sex determination mechanisms and mating or spawning behaviour.

evolution of sequential hermaphroditism based on the relationship between size and fecundity[30,32–34]. Specifically, since most fish, including sequential hermaphrodites, have indeterminate growth[35], i.e., they can keep growing as far as food resources and environment allow, sex change should be favoured when the reproductive value of an individual depends on size (and thus age), and this affects in particular one of the sexes. Thus, individuals change from smaller first sex to larger second sex and the direction of sex change depends on the sex that maximises its reproductive value with a larger size[36]. The size advantage model has been supported in crustaceans[37], molluscs[38], and teleost fish[39].

The interdependence between size, fecundity and fitness is affected by a species' mating system, defined as the pattern of sexual interactions given the number of reproducing males and females (Table 1). Therefore, among sequential hermaphrodites, protandry (male-to-female sex change) is usually expected in species that reproduce in monogamous or random pairs and where individuals switch from small males to large, highly fecund females, achieving higher reproductive potential. Conversely, protogyny (female-to-male sex change) is usually expected in polygynous/group-mating species, where small females become large dominant males that monopolise females, often grouped in harems (Table 1). In both systems, cases exist with a few individuals born directly as the second sex. Specifically, in digynic protandrous species, primary females directly develop as such and secondary females develop from males after sex change[40]. Likewise, in diandric protogynous species, primary males develop directly as such whereas secondary males develop from females after sex change[41].

Crucially, life-history traits underpin the formulation and assumptions of the size advantage model. Life history theory is central to the study of sexual systems evolution since it allows to derive clear predictions about why and when individuals should allocate energy among different life-history traits, including

sexual functions, to optimise fitness[42]. However, life-history traits are surprisingly not explicitly and formally incorporated in the size advantage model, nor tested in empirical studies[13]. Longevity, maximum size and age/size at first maturity are key life-history traits because they determine individual fitness, influence demographic parameters of populations[43] and impact populations' genetic diversity[44]. These traits evolve and are under several selective forces at the population level, but differences in the intensity of selection among species can lead to large-scale diversity, thus allowing large-scale comparative studies to inform our understanding of how and why they evolved[45]. Since sequential hermaphrodites achieve higher fitness when reproducing as the second sex[36], hence the advantage of changing sex, they should, on average, benefit more than gonochoristic and simultaneous hermaphroditic species from increased longevity (overall and/or as the second sex in particular), or larger size, especially in protandry, where females are the larger sex as size gives fecundity advantage. In general, larger females tend to produce more eggs than smaller ones both within and across species[46], while larger males do not necessarily increase their sperm production with size. In males, larger size gives the advantage to secure dominance and increase fertilisation rates, but not necessarily fecundity. Alternatively, sequential hermaphrodites could mature, on average, earlier as the first sex compared to the same sex in gonochoristic species and capitalise on reproduction as the second sex. These predictions, however, remain to be tested.

Although exceptions occur, spawning behaviour, i.e., how the two sexes interact to release the gametes, can be broadly classified in fish as pair spawning, involving only two individuals at the time, and group spawning, comprising large breeding groups[47] (Table 1). Mating system and spawning behaviour together determine the intensity of direct male-male competition and sperm competition (i.e., the competition between the sperm of two or more males for fertilisation of the same eggs), and thus the

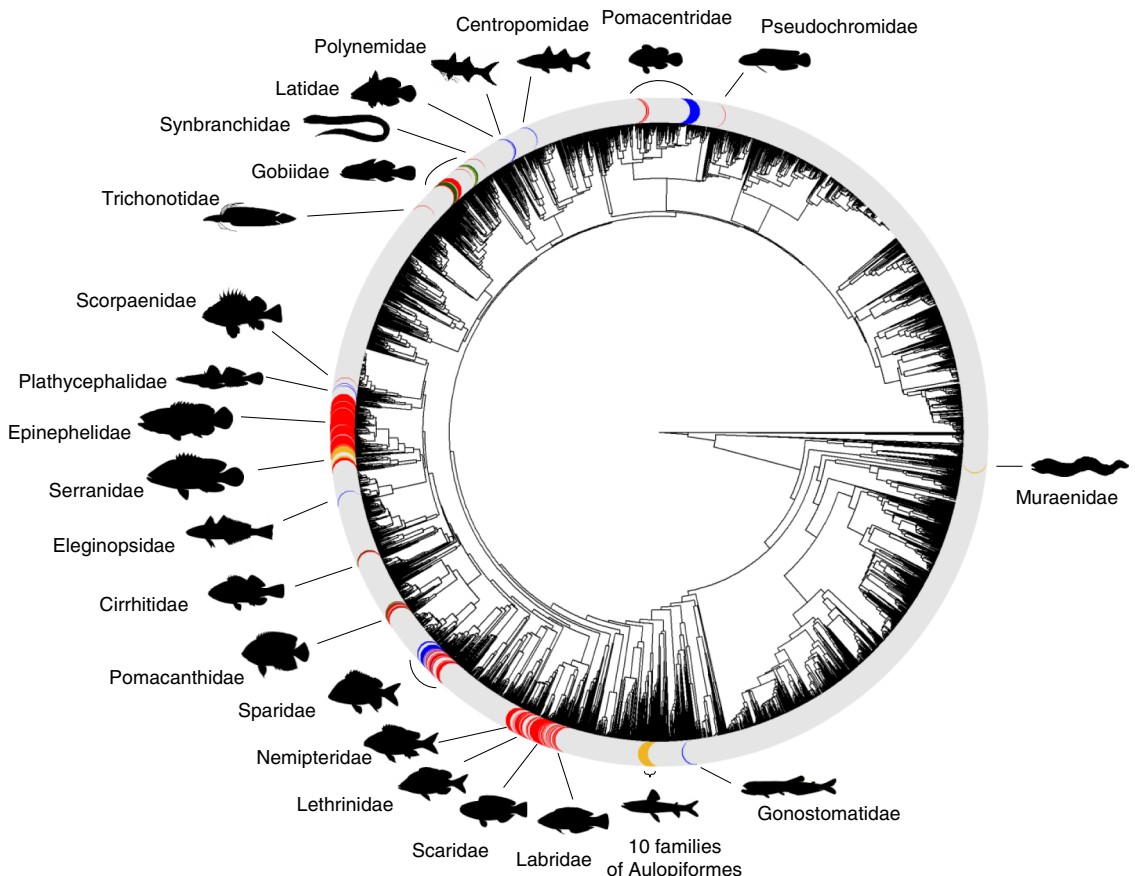

**Fig. 2 Sexual systems of extant species of teleosts.** Sexual systems are colour coded for gonochorism (n = 4320; grey), protogyny (n = 196; red), protandry (n = 36; blue), bidirectional sex change (n = 16; green) and simultaneous hermaphroditism (n = 46; yellow). Families (n = 32) with hermaphroditic species are labelled. Silhouettes have been obtained from fishualize[108] or drawn by one of the authors (C.B.).

certainty of paternity. Sperm competition is a key selective force shaping male reproductive anatomy, physiology and behaviour across diverse animal groups[48,49]. In general, sperm competition is assumed to be low in haremic systems[39,50,51] where large dominant males can better monopolise groups of females (pair spawning) with no or limited competition by other males[52]. Likewise, low sperm competition is expected under monogamy. Group spawning is commonly found in promiscuous mating, leading to intense sperm competition[53] as many males try to fertilise the eggs of multiple females. The intensity of sperm competition has been incorporated in the size advantage model[54] as it can play a significant role in the advantage of protogyny: changing sex from female-to-male should be more advantageous when paternity assurance is high due to reduced sperm competition[55]. Consistent with these predictions, the gonadosomatic index (GSI), defined as the percentage of body mass devoted to the gonads[56] and a reliable indicator of the intensity of sperm competition[57], is significantly lower in protogynous teleost species than in gonochoristic congeners[51,52,58]. However, protandrous teleost fish do not always conform to theoretical expectations, exhibiting higher GSI as males than expected[52]. We have recently proposed that, at least in the family Sparidae, high male GSI in protandrous fish can be explained not only by group spawning and high sperm competition in some species but also because high investment in the gonads can represent a compensatory mechanism that allows small males to fertilise highly fecund females much larger than themselves[58].

Teleosts account for more than 50% of the extant species of vertebrates and are characterised not only by their extraordinary diversity in morphology, physiology, ecology and habitat but also by different sexual systems, including gonochorism, different forms of hermaphroditism —the only group among vertebrates— and unisexuality (all-female populations)[11,59,60]. Hermaphroditism in teleosts is broadly divided into simultaneous (synchronous) and sequential (consecutive) hermaphroditism, the latter in the form of protandry, protogyny and bidirectional sex change (Table 1). Thus, the remarkable diversity in sexual systems in teleost fish makes them an ideal group in which to study the evolution of different forms of hermaphroditism[29]. Here, we investigate the evolutionary origin and transitions among sexual systems across 4614 teleost species belonging to 49 orders and 293 families using a recent time-calibrated phylogeny[61] and modern phylogenetic comparative approaches. Our large-scale approach allows us to fully unravel how sexual patterns evolved and identify which ones represent evolutionary stable conditions. We focus on gonochorism, protogyny, protandry and simultaneous hermaphroditism as these are the most common sexual systems in teleosts. For hermaphrodites, we only included species for which functional hermaphroditism could be confirmed by primary literature; all remaining species, following the sexual system obtained from FishBase[62], were classified as gonochoristic, excluding the species with ambiguous information (see Methods and Supplementary Fig. 1). We do not distinguish digynic and diandric species (or populations) in this study because the number of sequentially hermaphroditic species in our dataset is not sufficient for splitting them into narrower categories. Thus, separating digynic and diandric species would lead to a small sample size per category while increasing the number of parameters to be estimated, ultimately

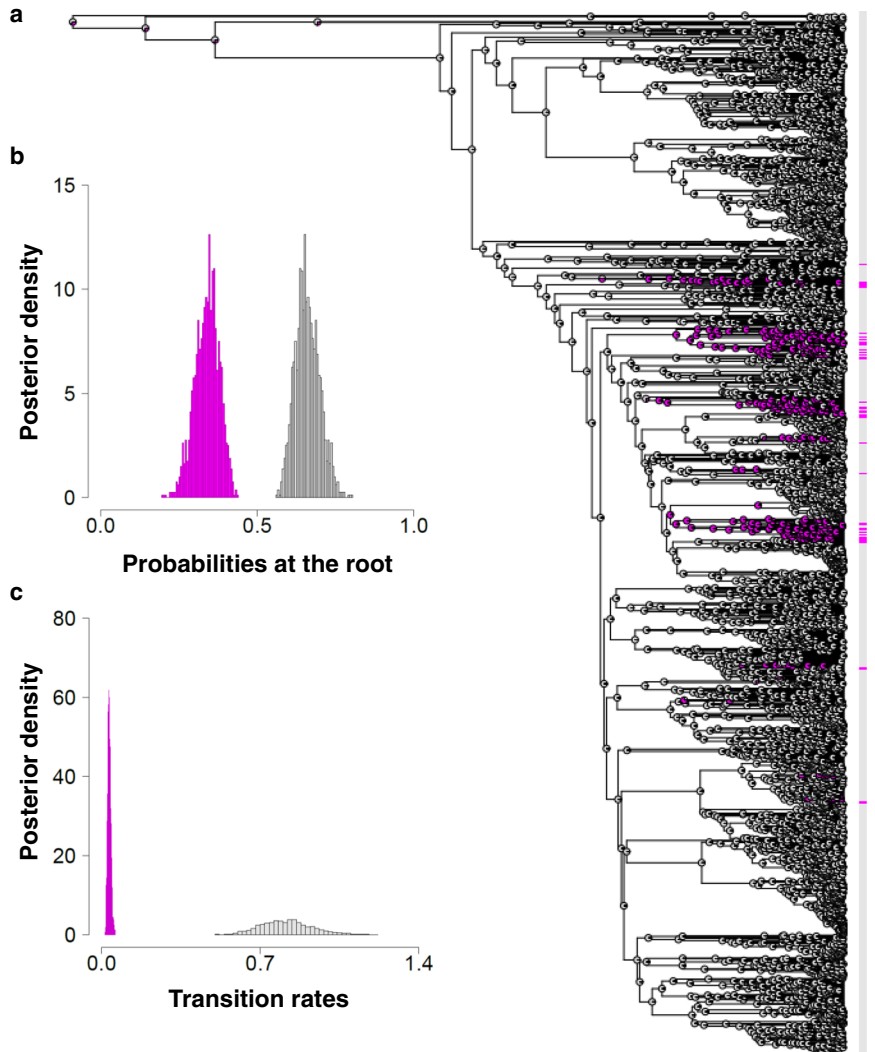

**Fig. 3 The evolutionary history of the sexual system in teleosts. a** Visual summary of maximum likelihood ancestral state reconstruction as a two-character state (gonochorism or hermaphroditism) that best approximates results of our RJ-MCMC Multistate model. The sexual systems of extant species and their ancestors are colour coded for gonochorism (*n* = 4320; grey) and hermaphroditism (*n* = 294; magenta). **b** Density plots from RJ-MCMC Multistate models for the estimated probability of character state at the root of the phylogeny colour coded for gonochorism (mean = 66%; grey) and hermaphroditism (mean: 34%; magenta). **c** RJ-MCMC multistate posterior distributions of the transition rates from gonochorism to hermaphroditism (magenta) and from hermaphroditism to gonochorism (grey).

eroding the power of the analysis. Likewise, unisexual species ("biotypes", hybrid in origin)[59,63] are too few to be incorporated in any formal analyses in our study.

Here, we demonstrate that gonochorism is the likely ancestral condition in teleosts and it is an evolutionarily stable state from which protogyny and protandry evolve at a moderate evolutionary rate. Consistent with theoretical predictions, we show that simultaneous hermaphroditism cannot evolve directly from gonochorism but rather through the intermediate stage of sequential hermaphroditism, most likely protandry. Further, we expand the theoretical framework of the evolution of hermaphroditism investigating how life-history traits and male GSI differ between sexual systems, as predicted by life history theory. In support of these predictions, we found evidence of a longer lifespan in protogynous species compared to gonochoristic and strong evidence of smaller GSI in protogynous males. However, contrary to predictions, we found no difference in maximum size and age and size at maturity across sexual systems. We discuss how our results should be incorporated into a broader framework with sex-determining mechanisms and gonadal plasticity as possible

constraining and facilitating mechanisms, respectively, to gain a fuller understanding of the evolution of sexual systems and possibly resolve Williams' paradox.

## Results

**Evolutionary history of sexual systems in teleosts**. Our dataset includes 4614 extant teleost species, of which 294 are hermaphroditic (protogynous: *n* = 196; protandrous: *n* = 36; bidirectional sex changers: *n* = 16; simultaneous hermaphrodites: *n* = 46; Fig. 2; Methods, Data collection and verification; Supplementary Data 1). We used Discrete models of evolution to reconstruct the evolutionary history of sexual systems using Reversible Jump (RJ) Markov chain Monte Carlo (MCMC) in *BayesTraits* (Methods, Phylogenetic comparative analysis). Treating sexual systems as a two-character state (gonochoristic or hermaphroditic) our analysis reveals that gonochorism is the most likely ancestral character state in teleosts (Fig. 3a, b; Supplementary Table 1) and that hermaphroditism evolves slowly from, and reverts very quickly and multiple times back to, gonochorism (Fig. 3a, c;

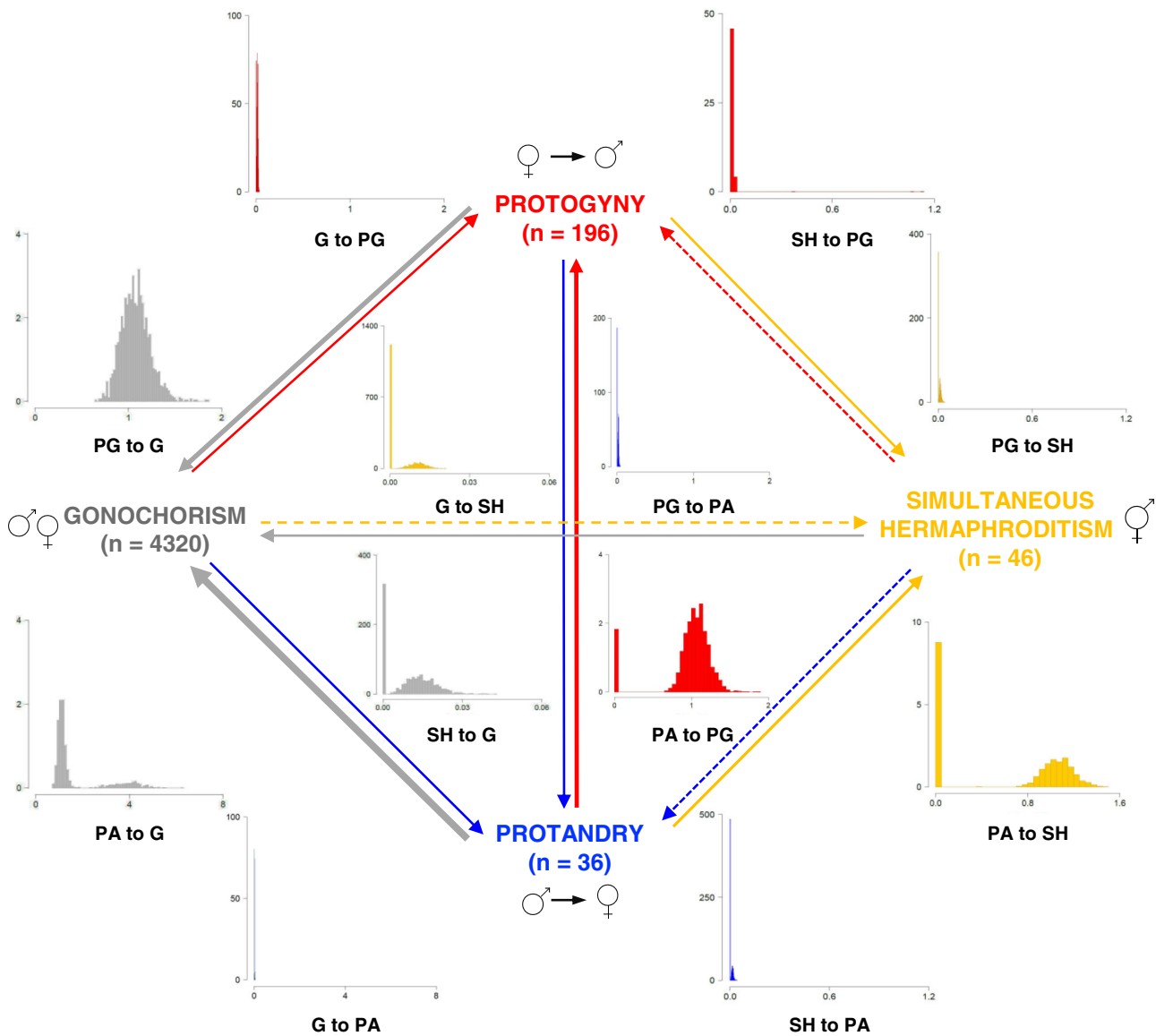

**Fig. 4 Transitions rates between sexual systems in teleosts.** Summary of RJ-MCMC Multistate analysis with density plots of the posterior distributions of the transition rates to gonochorism (grey), protogyny (red), protandry (blue), and simultaneous hermaphroditism (yellow). Gonochorism is the estimated likely ancestral condition. Note, only x axis, but not y axis, are the same for each pair of gain and loss between two-character states. The thickness of the arrows is roughly proportional to the mean magnitude of the transition rates from the posterior distribution. Dashed arrows indicate transition rates estimated to be equal to 0 in over 40% of the models in the posterior distributions. Sample sizes of extant species included in our analysis for each sexual system category are indicated between parentheses.

Supplementary Table 1). This indicates that gonochorism is an evolutionarily stable state in teleosts.

Treating the sexual system as a four-character state (gonochoristic, protandric, protogynic, simultaneous hermaphroditic; Fig. 2) reveals that both types of sequential hermaphroditism evolve at a very low rate from gonochorism and revert very rapidly back to it (Fig. 4, Table 2). In contrast, direct transitions between gonochorism and simultaneous hermaphroditism are very slow if they happen at all, given that over 60% and 31% of the models estimate the transition from gonochorism to simultaneous hermaphroditism and the reversal, respectively, to be equal to zero. Our analysis also shows that protogyny evolves as slowly from gonochorism as it switches to protandry and simultaneous hermaphroditism, although 32% of models estimate the latter transition to be equal to zero. Conversely, protandry is lost quickly to protogyny and simultaneous hermaphroditism, and

very rapidly to gonochorism. Therefore, simultaneous hermaphroditism evolves from sequential hermaphroditism, most likely from protandry, and is lost to gonochorism, protogyny and protandry at similarly low rates, although approximately half of the models estimate transition rates to sequential hermaphroditism to be equal to zero. Altogether, these results clearly indicate that in teleosts gonochorism is an evolutionarily stable state; protogyny is evolutionarily more stable than protandry, while simultaneous hermaphroditism evolves rarely, most like from protandry, and is evolutionarily stable being lost slowly to gonochorism, and less likely, to protogyny and protandry (Fig. 4, Table 2).

**Life-history traits and sexual systems in teleosts.** Using life-history theory, we predicted that sequential hermaphrodites live

**Table 2 Results of the RJ-MCMC Multistate analysis in *BayesTraits* of sexual systems as a four-state categorical variable: gonochorism (G), protogyny (PG), protandry (PA), or simultaneous hermaphroditism (SH).**

| Transition rates | ESS | Mean | 95-HPD | Mode | % Zero |
|---|---|---|---|---|---|
| G → PG | 1153 | 0.014 | 0.000–0.023 | 0.014 | 7.4 |
| PG → G | 1600 | 1.077 | 0.804–1.396 | 1.116 | 0.0 |
| G → PA | 1600 | 0.015 | 0.005–0.025 | 0.014 | 0.0 |
| PA → G | 1143 | 1.617 | 0.760–4.292 | 1.105 | 0.0 |
| G → SH | 1366 | 0.004 | 0.000–0.014 | 0.000 | 60.7 |
| SH → G | 1600 | 0.010 | 0.000–0.023 | 0.000 | 31.7 |
| PG → PA | 1600 | 0.013 | 0.000–0.023 | 0.014 | 18.7 |
| PA → PG | 1600 | 0.976 | 0.000–1.321 | 1.107 | 2.9 |
| PG → SH | 1600 | 0.009 | 0.000–0.023 | 0.000 | 35.8 |
| SH → PG | 1600 | 0.009 | 0.000–0.021 | 0.000 | 51.7 |
| PA → SH | 1155 | 0.602 | 0.000–1.236 | 1.105 | 18.1 |
| SH → PA | 1600 | 0.008 | 0.000–0.021 | 0.000 | 48.6 |
| Root probabilities | | | | | |
| G | 1324 | 46.1 | 36.1–55.0 | 49.5 | 0.0 |
| PG | 1258 | 31.4 | 22.7–37.8 | 34.9 | 0.0 |
| PA | 1167 | 22.4 | 13.1–37.0 | 14.9 | 0.0 |
| SH | 1600 | < 0.1 | 0.0– < 0.1 | 0 | 6.8 |

For each posterior distribution, we report the effective sample size (ESS), the mean and 95% high posterior density intervals (95-HPD), the mode and the percentage of models in which the parameter is estimated as zero. This analysis is based on 4598 extant species (G: n = 4320; PG: n = 196; PA: n = 36; SH: n = 46). Note: 16 species of bidirectional sex change were not included in this analysis due to their low number.

longer and/or reach larger adult size and/or mature earlier as the first sex than gonochoristic species. The phylogenetic generalised least square (PGLS) analyses revealed that protogynous, but not protandrous, species live longer than gonochoristic species (Fig. 5a; Table 3). Larger species however might live longer, therefore we repeated the analysis controlling for allometry; even so, adding size (maximum length) as a covariate did not alter this result (Fig. 5b; Supplementary Table 2). Contrary to predictions, we did not find any significant size difference across sexual systems (Fig. 5c; Table 3). Female and male age at first maturity does not differ across species with different sexual systems (Fig. 5d, e; Table 3), even when accounting for allometry (length at maturity; Supplementary Table 4), nor does sex-specific length at maturity (Table 3). Finally, the PGLS revealed that protogynous males have lower GSI values than gonochoristic and protandrous ones, but GSI does not differ significantly between gonochoristic and protandric males (Fig. 5f; Table 3) even when considering allometry (Supplementary Table 2).

## Discussion

Our large-scale phylogenetic study has tested the theoretical predictions on how sexual systems evolve and has revealed the evolutionary origin of and transitions between different sexual systems in the highly diverse teleosts. We identify gonochorism and simultaneous hermaphroditism as stable conditions over evolutionary time. In support of recent theoretical models[4], our study demonstrates that simultaneous hermaphroditism is unlikely to evolve directly from gonochorism and instead requires the intermediate step of sequential hermaphroditism, most likely protandry. We find support for the predictions derived from the life-history theory that protogynous species live longer than gonochoristic species but no evidence that sequential hermaphrodites attain a larger size or mature earlier than gonochoristic species. Finally, we find strong evidence that protogynous males invest the least in male gonad tissues (quantified by the gonadosomatic index) relative to gonochoristic and protandric males.

Combined, these results suggest that the two forms of sequential hermaphroditism must be treated separately in theoretical and empirical studies as protandry and protogyny are characterised by distinct life history strategies[36], even though they both entail sex change.

Using the largest dataset ever collected with four sexual systems in teleosts, our study reveals a complex and dynamic way through which sexual systems evolve and switch between one another. Sequential hermaphroditism can evolve slowly from gonochorism, the ancestral state in teleosts, but revert to gonochorism rapidly. Although gonochorism is an evolutionarily stable condition, gained faster than it is lost, these results refute the assumption that the transition to gonochorism is irreversible[20] and represent another example[64,65] against Dollo's law of irreversibility[66], as previously suggested[29]. Conversely, sequential hermaphroditism in teleosts, particularly protandry, is less evolutionarily stable than gonochorism. Our results however contradict Pennell et al.'s[29] finding that the evolutionary transition from gonochorism to hermaphroditism occurs over twice as fast as the reverse, suggesting rapid evolution of hermaphroditism from gonochorism, a conclusion that the same authors acknowledge is counterintuitive. In contrast, we find that the evolutionary gain of hermaphroditism is slower than its loss to gonochorism, regardless of whether we treat sexual system as a binary trait (gonochorism *vs* hermaphroditism) or discriminate between different forms of hermaphroditism. Heterogeneity in the rate of gain and losses across large phylogenies can potentially bias the estimates of the faster transition rate for binary traits[67]. However, our analysis of four states reveals that protandry is lost rapidly to both protogyny and gonochorism, and to a lesser degree, to simultaneous hermaphroditism.

We note that Pennell et al.[29] used a much smaller dataset biased towards a greater proportion of hermaphroditic than gonochoristic species than what is observed in teleosts and did not discriminate between different types of hermaphroditism. We have accepted the classification in FishBase[62] for gonochoristic species (unless rejected or disputed by primary literature), without individually confirming their sexual system as done for the hermaphroditic species in our dataset. This is because gonochorism is rarely confirmed in primary sources even when present in fish. As a result, if we used only a few gonochoristic species for which sexual system is explicitly confirmed in the original sources, the dataset would be strongly biased against gonochorism and include an unrealistic small number of gonochoristic species, ultimately undermining the robustness of the results. However, we acknowledge that a few species currently classified as gonochoristic in our dataset might be hermaphroditic. Although it is not possible to predict how this could influence the outcome of the analysis, given that this depends on the number of affected species, their phylogenetic position and the sexual system of their closely related species, our results represent an accurate picture of the evolution of the sexual system in fish with the data currently available. Crucially, our results at four states indicate that rapid transition rates from hermaphroditism to gonochorism in our analysis at two states are robust and reveal that protandry and protogyny—but not simultaneous hermaphroditism—evolve much more slowly from gonochorism than the reverse.

Importantly, our study demonstrates that simultaneous hermaphroditism does not originate directly from gonochorism but rather through sequential hermaphroditism, most likely protandry. However, simultaneous hermaphroditism is lost preferentially to gonochorism than to either form of sequential hermaphroditism. Thus, our analyses demonstrate that an intermediate stage is required for the gain of simultaneous hermaphroditism from gonochorism but not the loss back to it.

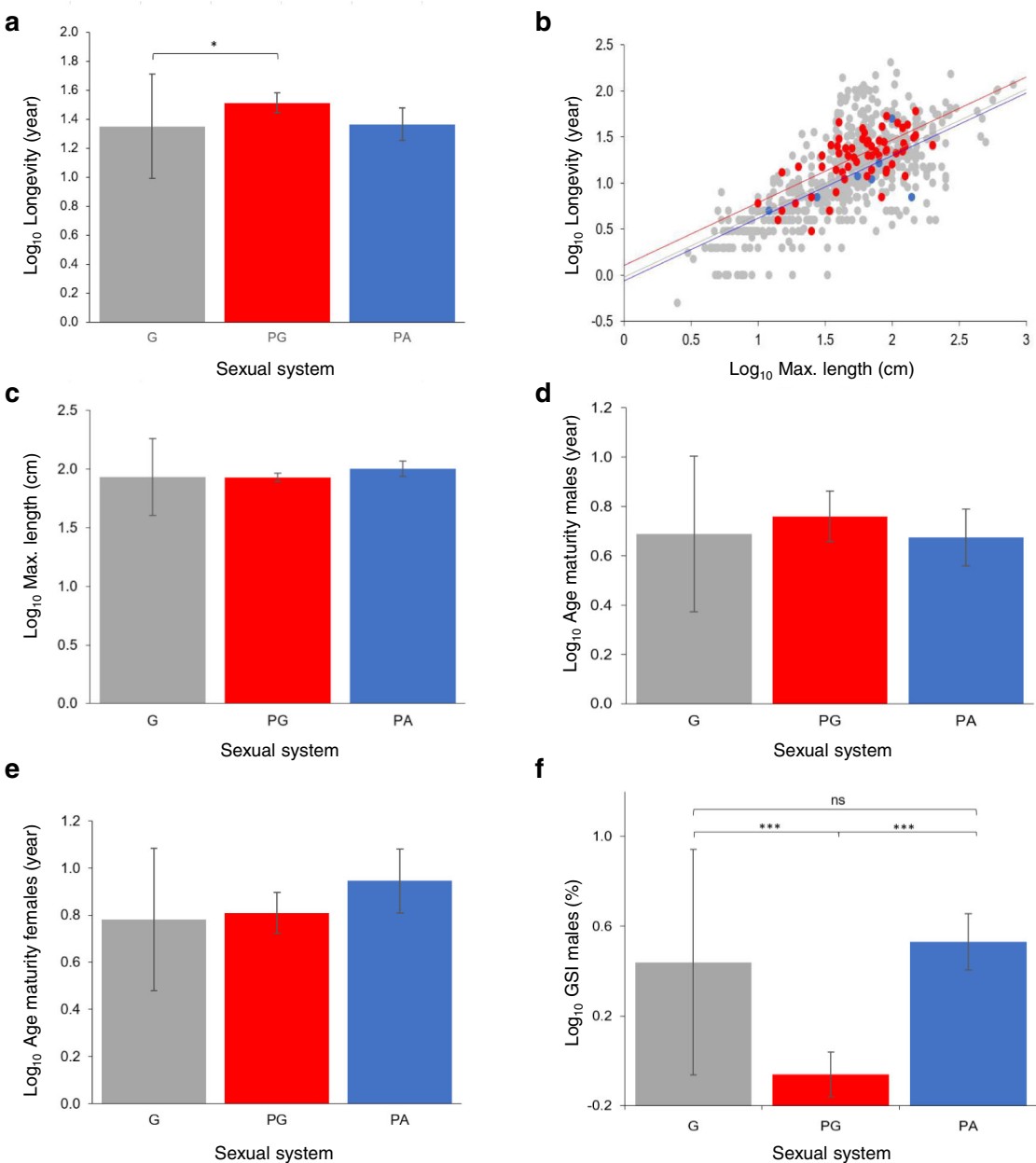

**Fig. 5 Life history traits by sexual system in teleosts.** Phylogenetic estimated mean and phylogenetic standard error from the PGLS results of: **a** longevity (year, $\log_{10}$ -transformed; G: $n = 758$; PG: $n = 69$; PA: $n = 17$); **b** longevity while controlling for maximum length (G: $n = 575$; PG: $n = 61$; PA: $n = 8$); **c** maximum length (cm, $\log_{10}$ -transformed; G: $n = 2612$; PG: $n = 167$; PA: $n = 20$); **d** male age at first maturity (year, $\log_{10}$ -transformed; G: $n = 259$; PG: $n = 15$; PA: $n = 9$); **e** female age at first maturity (year, $\log_{10}$ -transformed; G: $n = 282$; PG: $n = 30$; PA: $n = 5$); **f** male gonadosomatic index, GSI ($\log_{10}$ -transformed; G: $n = 44$; PG: $n = 38$; PA: $n = 15$). In all panels gonochorism (G) is depicted in grey, protogyny (PG) in red and protandry (PA) in blue. *$P < 0.05$; ***$P < 0.001$ (please refer to Table 3 for details). Source data are provided as a Source Data file.

These results support theoretical predictions (Fig. 1) that sex-specific gene expression in gonochoristic species may prevent direct evolutionary transitions between gonochorism and simultaneous hermaphroditism, and intermediate stages, like sequential hermaphroditism, are required[4]. Overall, our study is consistent with suggestions that the complexity of sex-specific physiology and behaviour is likely to constrain some transitions between sexual systems. Androdioecy is considered an intermediate stage from simultaneous hermaphroditism to dioecy in plants and from gonochorism to hermaphroditism in some invertebrates[5,14]. However, this sexual system is extremely rare in fish and cannot explain the evolution of the diverse sexual systems in this vertebrate group, where instead sequential

hermaphroditism seems to play a similar role. We suggest that future studies in other taxa may also consider sequential hermaphroditism (if present) together with other mixed systems as an important stepping stone for evolutionary changes between gonochorism and simultaneous hermaphroditism as we have found in teleosts.

According to life-history theory, sequential hermaphrodites should, on average, live longer, grow bigger and/or mature earlier as the first sex than gonochoristic species. We find that protogynous and protandrous species differ in their life history strategies: protogynous, but not protandrous species, live longer than gonochoristic species. These differences reflect the fact that protogyny and protandry maximise their fitness as the second sex[36]

**Table 3 Results of phylogenetic generalised least square (PGLS) model of longevity (year; $log_{10}$ -transformed), maximum length (cm; $log_{10}$ -transformed), age at first maturity (year; $log_{10}$ -transformed), length at first maturity (cm; $log_{10}$ -transformed) per each sex (♂: male; ♀: female), and male gonadosomatic index (GSI; $log_{10}$ -transformed) across sexual systems: gonochorism (G); protogyny (PG); protandry (PA).**

| Variable | | Beta | T | P | Df | Model statistics | |
|---|---|---|---|---|---|---|---|
| **Dependent** | **Independent** | | | | | **λ** | **$R^2$** |
| Longevity | Sexual system – PA[1] | 0.015 | 0.128 | 0.898 | 2; 841 | 0.914 | 0.007 |
| | Sexual system – PG[1] | 0.161 | 2.340 | **0.019** | | | |
| | Sexual system – PG[2] | 0.146 | 1.205 | 0.229 | | | |
| Max length | Sexual system – PA[1] | 0.068 | 1.036 | 0.300 | 2; 2796 | 0.972 | 0.0004 |
| | Sexual system – PG[1] | −0.036 | −0.098 | 0.922 | | | |
| | Sexual system – PG[2] | −0.071 | −0.971 | 0.332 | | | |
| Age at first maturity ♂ | Sexual system – PA[1] | −0.015 | −0.129 | 0.897 | 2; 280 | 0.859 | 0.002 |
| | Sexual system – PG[1] | 0.070 | 0.684 | 0.495 | | | |
| | Sexual system – PG[2] | 0.085 | 0.623 | 0.534 | | | |
| Age at first maturity ♀ | Sexual system – PA[1] | 0.165 | 1.217 | 0.225 | 2; 314 | 0.862 | 0.005 |
| | Sexual system – PG[1] | 0.029 | 0.333 | 0.739 | | | |
| | Sexual system – PG[2] | −0.135 | −0.945 | 0.345 | | | |
| Length at first maturity ♂ | Sexual system – PA[1] | −0.060 | −0.809 | 0.419 | 2; 359 | 0.974 | 0.002 |
| | Sexual system – PG[1] | −0.020 | −0.337 | 0.736 | | | |
| | Sexual system – PG[2] | 0.040 | 0.448 | 0.654 | | | |
| Length at first maturity ♀ | Sexual system – PA[1] | −0.018 | −0.169 | 0.866 | 2; 340 | 0.971 | 0.0009 |
| | Sexual system – PG[1] | −0.041 | −0.565 | 0.572 | | | |
| | Sexual system – PG[2] | −0.023 | −0.200 | 0.842 | | | |
| GSI ♂ | Sexual system – PA[1] | 0.092 | 0.736 | 0.464 | 2; 94 | 0.835 | 0.234 |
| | Sexual system – PG[1] | −0.500 | −4.977 | **$2.920^{-06}$** | | | |
| | Sexual system – PG[2] | −0.592 | −4.209 | **$5.862^{-05}$** | | | |

[1]G as reference level.
[2]PA as reference level.
For each independent variable, we report the parameter estimate (Beta), t-statistics (T), P value (P; two-sided test), and the model statistics including the degrees of freedom (df), the maximum likelihood estimation of the phylogenetic signal (λ) and $R^2$. Significant differences are indicated in bold. Results of analyses controlling for allometry are available in Supplementary Table 2. See Supplementary Table 3 for sexual system and sex-specific sample sizes.

which differs between the two systems. Therefore, the longer life in protogynous species favours large successful males (second sex) that can monopolise females in harems or in spawning grounds. Conversely, protandrous species benefit primarily by achieving a larger size, as larger females (second sex) are more fecund than smaller ones. In addition, male investment in gonad tissue (as quantified by the gonadosomatic index) is lower in protogyny, as expected by theory[52,58], since large males can better monopolise mating opportunities and face low levels of sperm competition in harems and group spawning (Table 1). Small-sized protandrous males in group spawning instead need to boost their investment in the gonads but even in the absence of sperm competition (monogamy) they require large gonads to fertilise highly fecund females, larger than themselves[58]. Thus, sexual systems and mating strategies affect life-history traits differentially in protogynous and protandrous species. It is well known that in sequential hermaphrodites the second sex always matures later and is larger than the first sex, so it is not surprising that in protandrous species females are significantly larger than males when reaching maturity, while in protogynous species males are significantly larger than females[55] (excluding the cases of primary females and primary males, respectively). Yet, no comparison has been made for size/age at first maturity for males and females across sexual systems. Life-history theory predicts that the first sex of sequential hermaphrodites matures earlier than the same sex in gonochorism, but, with the data currently available, we find no evidence for this.

Our study includes explicitly life-history traits into a theoretical framework for the evolution of sexual systems and provides some evidence in support of theoretical predictions. However, records on life-history traits for teleosts species in general, and hermaphroditic species in particular, are still scarce. Even less complete and reliable data are available on mating systems and spawning behaviours, which should be incorporated in future studies aiming at obtaining a more complete understanding of the role that life-history traits play. Particularly necessary to fully assess theoretical predictions are sex-specific data for size and time spent as females and as males in sequential hermaphrodites, and for investment in male vs female function in simultaneous hermaphrodites, for which currently little is known. Future studies should re-evaluate these relationships as more data become available for a large number of species. While we have shown that life history theory can provide a major contribution to our understanding of sexual system evolution, below we present a general model for studying sexual systems and propose that the highly dynamic picture revealed by this study should be expanded using a more comprehensive approach that includes not only selection and adaptation, but also sex-determining mechanisms and gonadal plasticity (Fig. 6).

Sex determination in gonochoristic animals is determined either at fertilisation by different genetic mechanisms, including male (XX/XY) or female (ZW/ZZ) heterogamety with homomorphic or heteromorphic sex chromosomes and polygenic systems, or after conception by environmental factors, or by a combination of both[68,69]. Fish are characterised by an incredible diversity[70–72] and plasticity[29,73] of sex-determining mechanisms. Many fish do not have sex chromosomes[71]. When they are present, they might not always be clearly differentiated since sex-determining loci might not be easily identifiable[74] and in some cases, the sex can be determined by a change in a single nucleotide[71]. High turnover of sex chromosomes has also been detected in some fish lineages (e.g., sticklebacks[75,76]), including reversal to autosomes[29]. It has been suggested that fixed, strongly canalised, genetic sex determination (culminating in the

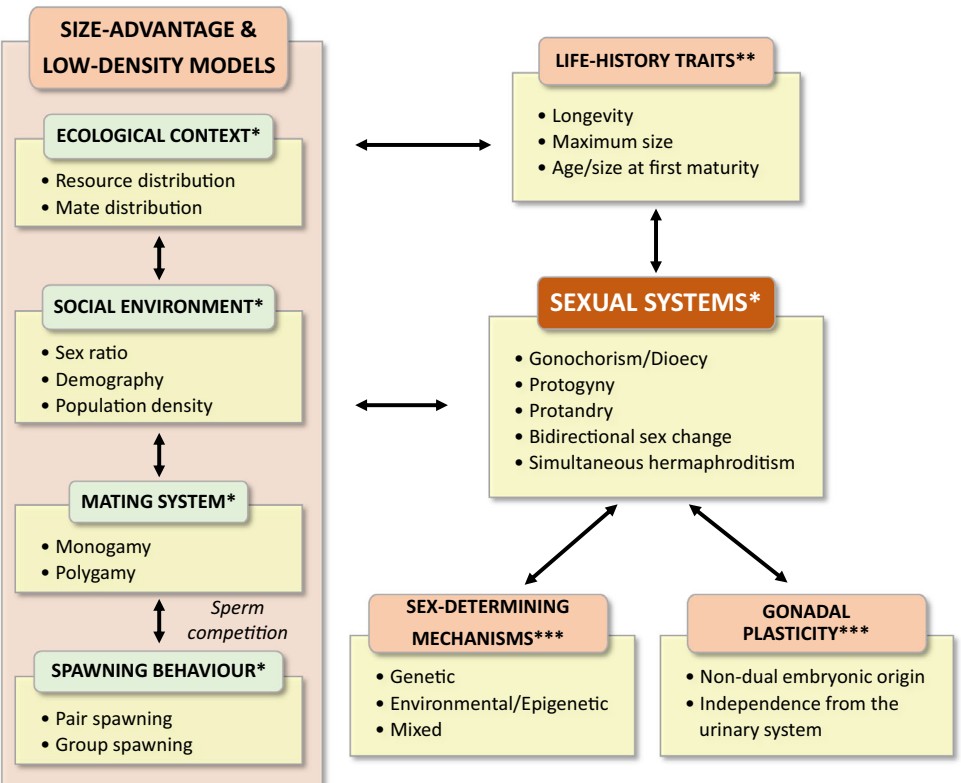

**Fig. 6 Theoretical framework for the study of the evolution of sexual systems in teleosts.** Overview of parameters (with some examples) considered in the low density and the size advantage models (*), used in our analyses (**) and proposed in the present study (***).

formation of fully differentiated and stable heteromorphic sex chromosomes) might constrain the evolution of hermaphroditism, acting as an evolutionary trap[2,77–79]. Even if this is not the case[29,80], sequential hermaphrodites do not appear to have sexually differentiated chromosomes[81], but data are currently scarce for formal analyses. Finally, sequential hermaphroditism can be regarded as a clear example of phenotypic plasticity, and since epigenetics underlies phenotypic plasticity, epigenetic mechanisms have been proposed to participate in the evolutionary transitions between different sexual systems and sex-determining mechanisms[82]. Therefore, although complete genetic control of hermaphroditism is common in plants[83], a better knowledge of the genetic and epigenetic mechanisms of sex determination could be helpful to explain how hermaphroditism in teleosts has evolved in some taxonomic groups but not in others under similar ecological pressures.

Previous attempts to connect the distribution of sexual systems have invoked morphological[84] and developmental[85] aspects. Developmental plasticity is uniquely documented in teleosts via the bipotential nature of their gonads and gonoducts[85]. Thus, while in most vertebrate taxa gonads develop from two distinct germinal layers (medulla, endodermal in origin, which gives rise to the testes; and cortex, mesodermal in origin, which gives rise to the ovary), in teleosts the gonads consist entirely of the cortex homologue[86]. Moreover, teleosts are the only group of vertebrates where the Müllerian duct is absent, and the gonoduct has the same origin in both sexes, being the reproductive system completely independent of the excretory system[85]. Therefore, anatomically all teleosts could, in principle, be hermaphrodites[85]. Furthermore, the transition to protogyny may be favoured by the peculiarities of gonadal development in many gonochoristic teleost species, which develop a female gonad, complete with ovaries containing cysts of oocytes, during the initial stage of gonadal

formation[87,88]. Only later testicular development is triggered and superimposed on this arrangement so that the individual ultimately matures functionally as a male[89–91]. Thus, protogyny might be favoured because female gonads are often the first to develop albeit temporarily, even in protandrous species[91]. Bidirectional sex change, a rarer system in teleosts (Table 1), further demonstrates the importance of gonadal plasticity. In most cases, the initial strategy is protogyny[92], but after sex change adult males can revert back to females when triggered by new social conditions. The retention of some female gonadal tissue in males facilitates a new change of sex, if and when required[93]. The maintenance of both gonadal tissues could facilitate a transition to simultaneous hermaphroditism. Thus, the study of the evolution of sexual systems in fish (and possibly other taxa) could greatly benefit from taking into consideration the facilitating/constraining aspects linked to gonadal developmental plasticity and the existence of different sex-determining mechanisms.

In conclusion, our study reveals that gonochorism is the most likely ancestral state and the most evolutionary stable sexual system in teleosts. In support of theoretical predictions, we demonstrate that simultaneous hermaphroditism cannot evolve directly from gonochorism but requires an intermediate step, most likely through protandry. However, simultaneous hermaphroditism is more likely to be lost to gonochorism than to sequential hermaphroditism in teleosts. Overall, our study reveals that the evolution of sexual systems is evolutionarily more dynamic and complex than commonly assumed. Our results support theoretical assumptions that changes between sexual systems are likely constrained by sex-specific gene expression, physiology and behaviour. In addition, we propose that the adaptive advantage of different sexual systems is further underpinned in fish by their extraordinary and unique developmental plasticity[94], including common and fast transitions among

different sex-determining mechanisms[29,68]. Our study also reveals that different sexual systems exhibit different life-history strategies that allow species with sequential hermaphroditism to maximise fitness as the second sex[36], particularly in protogynous species, and highlights the need for more sex-specific life-history data to gain a fuller and deeper understanding of the interplay between life-history strategies and sexual systems. Altogether we propose that a comprehensive framework that incorporates life-history traits, sex-determining mechanisms and gonadal plasticity into traditional theoretical models of sexual system adaptive value will be essential if we are to fully understand the evolution of sexual systems, their phylogenetic distribution and their implications for conservation and management.

## Methods

**Data collection and verification.** We compiled the most comprehensive database on sexual systems in teleosts to date. A dataset was first extracted from FishBase[62] for a total of 10914 actinopterygian species, of which 10875 were teleosts. Information on the sexual system was available for 9005 teleost species. Of these, 4740 species were included in the most recent and largest molecular phylogeny for the class[61] (available at https://fishtreeoflife.org; Supplementary Fig. 1). Next, species were classed as hermaphroditic only if functional hermaphroditism could be confirmed by primary literature, as recently compiled elsewhere[95], with further species confirmed from the primary literature. For the remaining species, we maintained the gonochoristic classification of FishBase[62], unless recent literature stated otherwise. Indeed, gonochorism is rarely confirmed in literature even when present, so including gonochoristic only species for which this sexual pattern is confirmed would strongly bias the dataset against gonochorism, ultimately undermining the robustness of the analyses. Importantly, species for which there is contrasting information in the literature were discarded. Altogether our final dataset included 4614 teleosts, with 4320 gonochoristic and 294 hermaphrodite species (Supplementary Fig. 1), of which there were 196 protogynous, 36 protandrous, 16 bidirectional species and 46 simultaneous hermaphrodites (Supplementary Data 1). Unisexual species were not included in the analyses, due to their extremely low number and hybrid origin[59,63]; we also did not have enough data (and power) to consider separately digynic and diandric species.

Life history traits (Supplementary Table 3) were also collected from the primary literature, FishBase[62] and *rFishBase*[96]: longevity (in years), maximum length (total length, TL in cm); age (in years) and length at first maturity (in cm) of males and females; male GSI (the maximum value recorded, expected to coincide with the peak of the reproductive season). When more than one value was present for longevity for a given species, we used the maximum value reported in the wild. We controlled for allometry as follows: longevity was controlled for maximum length (available for both sexes combined); age at first maturity was controlled for length at first maturity (by sex). GSI was controlled for male length at first maturity (male-specific): in this case, we could not use maximum length, not sex-specific, which would give an incorrect length of males in protandric species, where the larger individuals are females.

**Phylogenetic comparative analyses.** We investigated the evolutionary history of sexual systems of 4614 teleost species using Multistate models in *BayesTraits* V.3[97,98] in a Bayesian framework. Multistate estimates instantaneous transition rates between alternative character states of a single categorical variable (i.e., the rate of change between states along the branches of a phylogeny), based on a continuous-time Markov model of evolution for discrete traits[99,100]. A high transition rate from one state to another indicates that the first state changes rapidly to the second state over evolutionary time. Therefore, a character state is evolutionarily stable when it is lost more slowly than it is gained[101]. Multistate also produces posterior distributions of the ancestral character state at the root of the phylogeny. We scaled the tree by a default constant (mean of 0.1) in all analyses[101] and used an exponential prior whose mean was seeded from a uniform hyperprior ranging from 0 to 10 to reduce inherent uncertainty and biases of prior choice[98]. We ran all Multistate analyses with Reversible Jump (RJ) Markov chain Monte Carlo (MCMC) methods. MCMC samples models in direct proportion to their fit to the data, generating a posterior distribution of parameter estimates for each transition rate, and RJ sets some parameters equal to zero or equal to one another, thereby reducing model complexity and over-parametrization[97,98,100]. As a result, posterior distributions of parameter estimates may not be normal; we thus summarised results by presenting the mean and mode of the posterior distributions of each parameter estimate, 95% higher posterior density, and percentage of models with parameters estimated to be 0. We ran all MCMC chains for 320 million iterations in addition to half a million iterations, sampling every 200,000 iterations. All chains converged and showed good mixing as indicated by their effective sample sizes of 2000 and visual inspections of their traces in Tracer v1.6[102]. All analyses were run in triplicate and the three independent chains converged on very similar solutions, leading to qualitatively similar results. Here we present the results from the first chain. We ran RJ-MCMC Multistate analysis on the sexual system (Supplementary Table 4) as a binary state (gonochoristic or hermaphrodite) and as a four-state categorical variable (gonochorism, protandry, protogyny, simultaneous hermaphroditism). Bidirectional sequential hermaphrodites were excluded from the latter analysis as the sample size of extant species was too low so the chains failed to converge and mix properly when the sexual system was studied as a five-state categorical variable. For the analyses with a two-character state, we graphed the evolutionary history of sexual systems on the phylogeny using maximum likelihood (ML) in the R package *ape* v.5.3[103], which provided a reasonably close approximation of the RJ-MCMC Multistate results (this was not the case with the four-character state analysis).

We used phylogenetic generalised least square (PGLS) models[104–106] to test for the association of each life-history trait, entered as dependent variables, with sexual systems entered as an independent discrete variable with three possible states (gonochorism, protogyny, protandry), as not enough data were available for simultaneous hermaphroditic species and bidirectional sex changers. PGLS models were run with the R package *caper*[107] in Maximum Likelihood. The parameter λ of PGLS models quantifies the strength of the phylogenetic signal in the model residuals[104]. λ ranges between zero (there is no phylogenetic structure in the data) and one (the species share similarity in trait values directly proportional to their common evolutionary time, under the Brownian motion model of evolution[104,106]). Continuous variables were $\log_{10}$-transformed to meet assumptions of normality. The statistical tests in PGLS were two-tailed.

**Reporting summary.** Further information on research design is available in the Nature Research Reporting Summary linked to this article.

## Data availability

Data were extracted from primary literature and FishBase (www.fishbase.org); the phylogenetic tree[61] is available at https://fishtreeoflife.org. All data collected and analysed in this study are included in this published article and its Supplementary Information files. Source data are provided in this paper.

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

## Acknowledgements

This project was supported by Spanish Ministry of Science and Innovation Grant PID2019-108888RB-I00 to F.P. and the Santander Universities Travel Award Grant 2016/2017 to C.B. and S.P. that facilitated the collaboration among the three institutions involved in this study. I.C. was supported by NERC (grant no. NE/K013777/1). We acknowledge the funding of the Spanish government through the 'Severo Ochoa Centre of Excellence' accreditation (CEX2019-000928-S). We acknowledge Professor Stefano Mariani for helpful discussion in the first stages of the project and Professors Stephen Weeks and Manfred Schartl for valuable comments on the manuscript.

## Author contributions

F.P. conceived the study. S.P., C.B., I.C., and F.P. designed the study. S.P. collected data with assistance from C.B. Data analyses were performed by S.P., C.B., and I.C., S.P., C.B., I.C., and F.P. wrote the manuscript.

## Competing interests

The authors declare no competing interests.
