## [Peer Review File · Nature Communications]

Switches, stability and reversals in the evolutionary history of sexual systems in fishReviewers' Comments:

Reviewer #1:

Remarks to the Author:

Pla et al. conducted up to now the most comprehensive phylogenetic analyses of transitions among reproductive modes in teleost fishes and supplemented it with insights from life-history. The manuscript is impressive and I strongly believe that the paper will be a milestone in our understanding of evolution of animal sex systems. I have several questions and suggestions that might improve the precision of the manuscript:

- 1) You use the gonadosomatic index (GSI) in males, which is a simple ratio. The problem with ratios is that they can change with size, the allometry of GSI was discussed in several papers. Please, check that your analyses of GSI are not biased by allometry.
- 2) An inherent problem with similar phylogenetic analyses is that it is difficult to differentiate between transition rates and rate heterogeneity (see e.g. King and Lee, *Systematic Biology* 2015, 64: 532–554 for a discussion of the problem in the case of the reconstruction of squamate oviparity and viviparity). I think that it should be mentioned in the Discussion.
- 3) Line 292: "The phylogenetic generalized least square (PGLS) analysis revealed that protogynous, but not protandrous, species live longer than gonochoristic species... Protandrous species reach larger maximum adult size than protogynous species (Fig. 4b)". According to Figs 4a,b, there is a large overlap among categories, are the found differences biologically meaningful?
- 4) Line 296: "However, when accounting for allometry, males in protogynous systems mature later than males in protandrous systems." Was the allometry controlled in other tested relationships as well? How did you account for the allometry? By adding body mass into the PGLS model? Why it was not done in the other tested relationships? I would suggest to add body mass to all models and testing contribution of body mass and the tested variable based on the selection of the best model based on AICc.
- 5) Why asexual fish lineages were not included into the analyses? Asexuality is also a reproductive mode present in teleosts.

Minor points:

- 6) The authors might incorporate information that teleosts lost oviduct that can allow plasticity leading to evolution of hermaphroditism (Adolfi et al. *Intersex, Hermaphroditism, and Gonadal Plasticity in Vertebrates: Evolution of the Müllerian Duct and Amh/Amhr2 Signaling*. *Annu Rev Anim Biosci.* 2019, 15;7:149-172.
- 7) "Williams' paradox" - with all my respect to George Williams, I think that this term is not so universally known to be a keyword, also, please, explain this paradox a little bit more in the text.
- 8) The first sentence: "Eukaryotes" - do not capitalize the common name
- 9) Fig. 1: the green and blue arrow suggest directionality from the most common to the rarest state, however, in the evolutionary study I would expect more from the ancestral to the derived states. We know that the most common trait is not at all necessary the ancestral one.
- 10) There is a typo in the generic name *Kryptolebias*.
- 11) Line 144: "digynic protandrous species" - explain the term digynic
- 12) The analysis is restricted to teleost fish, but you use the colloquial term "fish" in many places. In fact, in the phylogenetic sense, all vertebrates are "fish". Importance of this distinction is best exemplified in the place where you mention a hagfish: mammals are more closely related to teleosts than hagfish, why you do not introduce sarcopterygians (lungfish, coelacanths, tetrapods) here? Please, use the cladistic system and define how you understand the term "fish" and better restrict your text to Teleostei.
- 13) Fig. 3: better do not use abbreviations like G, SH, PA... They make the figure more difficult to read.

Reviewer #2:

Remarks to the Author:

This paper has an interesting premise to explore the incidence of different forms of sexual patterns among the fishes in relation to phylogeny and selected life history traits.

While the overall approach is interesting, there are some fundamental questions that need to be addressed and points clarified before the analysis can be evaluated. There are core issues with mating system classification, database and predictions. Details are given as follows:

1. Mating system classification:

- a. The definitions of mating system used are unclear, appear confused and sometimes misleading or incorrect. Mating systems in fishes are more complex than indicated, an aspect of central relevance to this paper. For example, a fundamental division assigned by the authors is between 'group' and 'pair' spawning, with systems such as 'spawning aggregations' treated as group spawning. However, from a mating system perspective (and relevant to the paper) the distinctions are far more subtle, an aspect that is also reflected in male GSI (which the authors also use). For example, some groupers pair-spawn in harems (low GSI protogynous) that are sedentary. Some species migrate and pair-spawn within spawning aggregations (low GSI and protogynous), some species migrate and group-spawn in aggregations (high GSI and gonochore). All of this detail is relevant to the paper and but not reflected in any way.
- b. I cannot find how 'mating systems' were classified for each species in the Supplementary Information. This is needed as for all the other parameters used.
- c. In discussing sexual patterns, it is stated: 'In both systems exceptions exist with a few individuals born directly as second sex (primary females in digynic protandrous species³⁵ and primary males in diandric protogynous species³⁶)'. These are treated as exceptions and yet these more complex expressions of sexual pattern are of much interest. Diandry, for example, is common in at least one very large taxon, the Labridae. Of the two male types in diandric species, one has low GSI and pair-spawning following sex change, and one has high GSI and group-spawning and is functionally gonochoristic. These details should be important for this paper and yet *Thalassoma bifasciatum* (for example, a very well studied species in this regard) is simply treated along with all other protogynous species and many of its life history parameters are listed as NA in the SI, which is incorrect as all the relevant life history details are published. There are many such examples where life history information is in the literature but missing from the database.
- d. Table 1 – mating system definitions are confusing and even misleading; (i) gonochoristic species can have harems (ii) what does the double arrow mean in column two, for gonochore species or protogynous species, for example? (iii) should specify if 'sex ratio' is adult sex ratio (which presumably it is) (iv) I don't really understand the male/female symbols in the last column. For example, if species is biased to one or the other sex why are the sexes equal in number?

2. Database:

- a. Many of the 4741 species listed with their life history characters in the SI have an assumed sexual pattern indicated (from FishBase?) which has not been confirmed. Note that FishBase sometimes makes generic assumptions on certain life history features, such as sexual pattern, so these need to be checked species by species in the literature. A wealth of life history parameters relevant to this paper can be found in publications of the last few decades.
- b. There appear to be only 271 species in the SI for which mating system (gonochorism versus hermaphroditism type) is actually confirmed and some of these are incorrect and outdated. For example, published literature after 2000 for *Epinephelus polyphekadion* gives the needed life history parameters and shows gonochorism and not protogyny for this species. A full comprehensive updated literature review is needed to prepare the database for analysis.

3. Predictions:

- a. The basic premise of the paper is summarized as: 'Since sequential hermaphrodites achieve higher fitness when reproducing as the second sex³⁴, they should benefit more from increased longevity or

larger size than gonochoristic and simultaneous hermaphroditic species. Alternatively, sequential hermaphrodites could mature earlier as the first sex compared to the same sex in gonochoristic species and capitalize on reproduction as second sex. These predictions, however, remain to be tested.' I do not understand the basic premise here. It needs to be explained more clearly (not just by referring to another paper). For example, the selective pressure for sexual pattern is within species and not across species i.e. that under certain environmental circumstances or mating systems large size confers greater RS in males in protogynous species within that species than gonochorism. Features such as longevity or maximum size are constrained or determined by other factors within species. Moreover, for many fishes maximum length and size of maturation are correlated and these can vary according to environmental conditions (e.g
b. . within a species, in colder environments, growth is overall slower and that affects max size and maturation size).

4. Phylogeny

a. Phylogenetic relationships within the teleosts are poorly understood. It is not clear what system of phylogeny is applied in this paper. The most recent phylogeny for Actinopterygii fishes that I am aware of, not cited in this paper, is by Hughes et al., 2018 PNAS vol 115. Comprehensive phylogeny of ray-finned fishes.....available online.

Reviewer #3:

Remarks to the Author:

This manuscript presents a very nice comparative analysis of sexual systems in teleosts. The scope of the analysis far exceeds earlier studies and allows the authors to gain significant new insights into the controls of vertebrate sexual systems. I think these large scale evolutionary studies can be important for advancing a field because they provide the opportunity to examine widespread ideas that have been gained from case studies on smaller sets of species. However the current manuscript does not hit it out of the park for me in communicating the most significant findings. There are some really good ideas in the tables and figures but they are not yet polished to deliver a cohesive story. For example, fig 2 desperately needs a legend and more polish in general. Why are some lineages labelled? Why are some lineages represented with icons and not others? Why not color code the branches to indicate some information about the analysis?

There should be clear visual representation of the characters under analysis here. You might be able to incorporate rate information here as well. I liked figure 1 but then wondered why the color and icons here were not carried through to figs 3 and elsewhere. The problem here is that there are four character states and the possible transitions between them are complex. The hypothesis space is also complex so the authors really need to do a better job of bringing clarity to the analyses. I think color coded icons would work much better in fig 3 than letter abbreviations. The rate comparison doesn't really come through in table 2. I would try plotting multiple posteriors on the same graph--the table might go to the SI. Fig 4 should be boxplots rather than bar plots. Similarly I wonder if there is a way to visually represent the pgl results as a multipanel scatter plot with the significant relationships indicated. The full table might go to the SI.

I found two issues with the discussion. First the ideas of Pennell et al seem highly relevant to the paper and should be incorporated into the introduction as part of hypotheses to test. Right now this discussion of refuting their study comes as a surprise. The discussion spends a lot of space on it so I think you would be better served by incorporating this earlier. Second, I found the discussion to be somewhat long and unfocused. It feels like the authors are trying to say everything here. I did not find the presentation of a generalized model for studying sexual systems to be especially helpful and I wonder if it might be better to show in a review paper rather than this study. Subheadings might be useful as well.

Overall I think the study has potential but I don't feel like it quite hits the mark in its current version.

Some final small comments. I think the authors are too colloquial with the designations of fish lineages. It is fine be informal with the taxonomic designations after you have defined the major lineages but right now you talk about fish in an informal way at the start of the paper before injecting systematic terms. I think you need a short description of the taxonomic focus of the study and a justification of them.

I found it odd that the source phylogeny is not cited in l205. The tree files for that paper are distributed via a website so if the authors used that it would be also appropriate to cite J. Chang, D. L. Rabosky, S. A. Smith, An r package and online resource for macroevolutionary studies using the ray-finned fish tree of life. *Methods Ecol. Evol.*

Fig 2 caption. Citation to ape is not needed. A legend explaining color key and internal external circles is.

RESPONSE TO REVIEWER COMMENTS

We thank all reviewers for their useful suggestions that have helped us improve our study. Following the concerns raised by Reviewer 2 regarding our dataset, we have:

- 1) re-checked one by one all hermaphroditic species we had considered before and in the revised version we: a) include as hermaphrodite only those species for which we can confirm the presence of functional hermaphroditism supported by primary sources, b) have discarded any hermaphroditic species for which we could not find the original source or could not confirm the sexual system.
- 2) gathered new primary sources to further expand our dataset.

Thus, after discarding some species and including new ones following the criteria explained above, we now present results with our revised dataset of 4614 species in total, of which 294 are verified functional hermaphrodites (as opposed to 270 in the original submission).

Most of the conclusions of our study still hold although some results using this new dataset change slightly. Specifically, we still find that: i) both types of sequential hermaphroditism can evolve slowly from gonochorism and revert quickly to it; ii) protogyny is lost at low rate to protandry and simultaneous hermaphroditism; iii) protandry is not evolutionarily stable and is quickly lost to gonochorism, protogyny and, to a lesser degree, simultaneous hermaphroditism. However, we now find that simultaneous hermaphroditism does not evolve directly from gonochorism (as in our earlier submission) but via sequential hermaphroditism, most likely protandry, thus confirming theoretical predictions. In addition, simultaneous hermaphroditism is more likely to be lost to gonochorism than to either form of sequential hermaphroditism. Analyses on life history traits confirm that protogynous (but not protandrous) species live longer than gonochoristic and present the smallest GSI. With the new dataset we no longer find evidence that protandrous species are smaller and mature later than protogynous species – as we discuss in the manuscript, more sex-specific life history data need to be collected to fully evaluate the role of life histories in the evolution of hermaphroditism.

Reviewer #1 (Remarks to the Author):

Pla et al. conducted up to now the most comprehensive phylogenetic analyses of transitions among reproductive modes in teleost fishes and supplemented it with insights from life-history. The manuscript is impressive and I strongly believe that the paper will be a milestone in our understanding of evolution of animal sex systems. I have several questions and suggestions that might improve the precision of the manuscript:

We thank the reviewer for his/her overall very favorable opinion about our study.

1) You use the gonadosomatic index (GSI) in males, which is a simple ratio. The problem with ratios is that they can change with size, the allometry of GSI was discussed in several papers. Please, check that your analyses of GSI are not biased by allometry.

We thank the reviewer for this comment. We have followed the reviewer's suggestion and included size in this analysis. However, please note that it is not appropriate to use maximum length in analyses on male GSI because the records of maximum length do not report the sex of the individuals measured. As a result, using maximum length could provide an incorrect estimate of size for males of protandric species (where the larger individuals are females). To circumvent this problem, we have used male length at maturity, for which enough data exist, to control for allometry. Importantly, our conclusions remain unaltered with and without allometric control (see Supplementary Table 2). Furthermore, we note that that male length at maturity is not significant in the analysis of GSI.

2) An inherent problem with similar phylogenetic analyses is that it is difficult to differentiate between transition rates and rate heterogeneity (see e.g. King and Lee, *Systematic Biology* 2015, 64: 532–554 for a discussion of the problem in the case of the reconstruction of squamate oviparity and viviparity). I think that it should be mentioned in the Discussion.

A major advance of our study relative to earlier work is investigating the evolutionary history of sexual system while discriminating among its various forms. However, to allow comparison with previous studies, we also present results of analyses in which sexual system is treated as binary trait (gonochorism vs. hermaphroditism) and we find that the loss of hermaphroditism is faster than its gain. When treating sexual system at 4 states, we show that the fastest transitions are the loss of protogyny and protandry to gonochorism, the transition from protandry to protogyny and, to a lesser degree, from protandry to simultaneous hermaphroditism. This reveals a more complex and dynamic picture of the evolutionary history of sexual system than previously appreciated using only 2 states.

King & Lee (2015) show that in large scale analysis of *binary* traits, rate heterogeneity across the phylogeny is 'captured' by the fastest transition rate and this can lead to incorrect estimates for this transition rate. While this problem might potentially affect our analysis of sexual system as binary trait, it is unlikely that all our fastest transitions (protogyny to gonochorism, protandry to gonochorism, protandry to protogyny and protandry to simultaneous hermaphroditism), two of which are reversal to gonochorism, just 'capture' rate heterogeneity. This suggests that our conclusion that the rate of gain of hermaphroditism is lower than the rate of its loss is robust regardless of how sexual system is treated. However, we acknowledge that the reviewer has raised an important point and now discuss rate heterogeneity in our ms (Lines 399-403).

3) Line 292: "The phylogenetic generalized least square (PGLS) analysis revealed that protogynous, but not protandrous, species live longer than gonochoristic species... Protandrous species reach larger maximum adult size than protogynous species (Fig. 4b)". According to Figs 4a, b, there is a large overlap among categories, are the found differences biologically meaningful?

Our figure depicts the *phylogenetic* mean and associated *phylogenetic* SE of the mean as derived from the PGLS output. We note that gonochorism is the largest and most diverse category, thus it is not surprising that this category exhibits high variance and SE. Unfortunately, we currently lack data on life history traits, particularly sex-specific traits, for most hermaphroditic species. Thus, in our revised ms we discuss the importance of addressing this knowledge gap for the interpretation of our results and we point out that more life history data (and ideally sex-specific data) is needed for a larger number of hermaphroditic species (Lines 451-459).

4) Line 296: “However, when accounting for allometry, males in protogynous systems mature later than males in protandrous systems.” Was the allometry controlled in other tested relationships as well? How did you account for the allometry? By adding body mass into the PGLS model? Why it was not done in the other tested relationships? I would suggest to add body mass to all models and testing contribution of body mass and the tested variable based on the selection of the best model based on AICc.

Data on body mass is virtually non-existing for most fish species; however, length is more commonly taken as a measure of size and thus available for many more species in our dataset. Following the reviewer’s recommendation, we have now included maximum length, or length at maturity when appropriate, as covariates in the PGLS models for all traits tested (see Supplementary Table 2). Specifically, for age at maturity we used length at maturity (by sex) and, as mentioned above, for male GSI we have used length at maturity (as this was male specific, contrary to maximum length). We also controlled lifespan for allometry using maximum length although we note that controlling for allometry in this case could be a circular argument as many fish can grow indeterminately, hence not only older fish grow larger, but also larger fish are older. Because the different models (with and without allometric adjustment) have different number of observations (because fewer species have also data on length), we could not compare the models using AICc values. Instead, we now present both analyses (Table 3 and Supplementary Table 2).

5) Why asexual fish lineages were not included into the analyses? Asexuality is also a reproductive mode present in teleosts.

The reviewer is correct that asexuality is a reproductive mode present in teleosts. However, the number of known fish species that are unequivocally or likely to be parthenogenetic is very small and mainly of hybrid origin (“biotypes”, Vrijenhoek, 1994; Moore, 1984) preventing us from including this sexual system in our analyses. We have clarified this in the revised ms (Lines 235-236 and lines 548-549).

Minor points:

6) The authors might incorporate information that teleosts lost oviduct that can allow plasticity leading to evolution of hermaphroditism (Adolfi et al. Intersex, Hermaphroditism, and Gonadal Plasticity in Vertebrates: Evolution of the Müllerian Duct and Amh/Amhr2 Signalling. *Annu Rev Anim Biosci.* 2019, 15;7:149-172.

We thank the reviewer for pointing this out. The reference suggested was already cited in the original submission (ref. #68 now #84). However, explicit mention to the loss of the oviduct and the uncoupling of the reproductive and urinary systems was not present. In the revised version, we now highlight this important difference between fishes and other vertebrates, which underpins their sexual plasticity and likely allows the emergence of different forms of hermaphroditism (Lines 498-502).

7) “Williams’ paradox” - with all my respect to George Williams, I think that this term is not so universally known to be a keyword, also, please, explain this paradox a little bit more in the text.

Agreed. “Williams’ paradox” has been removed as a keyword and is now better explained in the revised manuscript (Lines 69-74).

8) The first sentence: “Eukaryotes” - do not capitalize the common name

The text has been corrected (Line 55).

9) Fig. 1: the green and blue arrow suggest directionality from the most common to the rarest state, however, in the evolutionary study I would expect more from the ancestral to the derived states. We know that the most common trait is not at all necessary the ancestral one.

We agree with the reviewer that the green and blue arrows were misleading. Also, the separation between animals and plants is not as strict (as in the case of androdioecy in animals), so we have removed the original directional arrows in updated Figure 1 altogether.

10) There is a typo in the generic name *Kryptolebias*.

Thank you for pointing this out. The typo has been corrected (Line 118).

11) Line 144: “digynic protandrous species” - explain the term digynic

Thanks for this suggestion. In the revised version we now better explain the term “digynic” as well as the term “diandric” (Lines 231-235 and lines 549-550).

12) The analysis is restricted to teleost fish, but you use the colloquial term “fish” in many places. In fact, in the phylogenetic sense, all vertebrates are “fish”. Importance of this distinction is best exemplified in the place where you mention a hagfish: mammals are more closely related to teleosts than hagfish, why you do not introduce sarcopterygians (lungfish, coelacanth, tetrapods) here? Please, use the cladistic system and define how you understand the term “fish” and better restrict your text to Teleostei.

We agree and, given that the focus of our paper is on teleosts, we have changed the terminology in the manuscript accordingly. We have left the example of the hagfish, however, as it represents an interested and still debated example of gynodioecy and trioecy (Lines 119-120).

13) Fig. 3: better do not use abbreviations like G, SH, PA... They make the figure more difficult to read.

Following this reviewer advice, in the new figure (now Figure. 4) the names of the different sexual systems are spelled out.

Reviewer #2 (Remarks to the Author):

This paper has an interesting premise to explore the incidence of different forms of sexual patterns among the fishes in relation to phylogeny and selected life history traits.

While the overall approach is interesting, there are some fundamental questions that need to be addressed and points clarified before the analysis can be evaluated. There are core issues with mating system classification, database and predictions. Details are given as follows:

We thank the reviewer for considering our study interesting and for the constructive criticism, which undoubtedly has helped to improve the ms. We have addressed all the concerns, as explained below.

1. Mating system classification:

a. The definitions of mating system used are unclear, appear confused and sometimes misleading or incorrect. Mating systems in fishes are more complex than indicated, an aspect of central relevance to this paper. For example, a fundamental division assigned by the authors is between ‘group’ and ‘pair’ spawning, with systems such as ‘spawning aggregations’ treated as group spawning. However, from a mating system perspective (and relevant to the paper) the distinctions are far more subtle, an aspect that is also reflected in male GSI (which the authors also use). For example, some groupers pair-spawn in harems (low GSI protogynous) that are sedentary. Some species migrate and pair-spawn within spawning aggregations (low GSI and protogynous), some species migrate and group-spawn in aggregations (high GSI and gonochore). All of this detail is relevant to the paper and but not reflected in any way.

We agree with the reviewer on the complexity of fishes’ mating systems and that our schematic classification does not allow the inclusion of unusual cases and exceptions (present in nature). We have clarified as much as possible the definitions of Mating system and Spawning behavior in Table 1 and the surrounding text citing it. Thus, Table 1 summarizes the main predictions derived from theory and applicable to most species. Thus, this table is suited to the large scale perspective of our study and it is not intended to incorporate predictions for species with unusual spawning or mating systems. However, we understand the reviewer’s concerns and in the revised ms we have reworded some sentences (using adverbs and wording like: on average, broadly, in general, usually, commonly, most common...) and added explicit definitions in the legend/footnote of Table 1, which clearly states that exceptions may occur in species with unusual spawning or mating systems.

b. I cannot find how ‘mating systems’ were classified for each species in the Supplementary Information. This is needed as for all the other parameters used.

We apologize for the confusion - unfortunately, the number of species with reliable information on mating system (and spawning behavior) is too small to formally include these variables in our analyses. Therefore, we only discuss these variables in the introduction given their relevance to the questions asked in our study and do so in general terms to focus on the broad patterns and theoretical predictions. We present in the SI only the data used in our analyses and we make the point in the Discussion that there is great need to collect these data so that they can be formally included in future studies (Lines 451-459 and see also Figure 6).

c. In discussing sexual patterns, it is stated: ‘In both systems exceptions exist with a few individuals born directly as second sex (primary females in digynic protandrous species and primary males in diandric protogynous species)’. These are treated as exceptions and yet these more complex expressions of sexual pattern are of much interest. Diandry, for example, is common in at least one very large taxon, the Labridae. Of the two male types in diandric species, one has low GSI and pair-spawning following sex change, and one has high GSI and group-spawning and is functionally gonochoristic. These details should be important for this paper and yet *Thalassoma bifasciatum* (for example, a very well-studied species in this regard) is simply treated along with all other protogynous species and many of its life history parameters are listed as NA in the SI, which is incorrect as all the relevant life history details are published. There are many such examples where life history information is in the literature but missing from the database.

This is a great point. Unfortunately, the number of protogynous and protandrous species in our dataset is insufficient to allow us to split them in narrower categories as this will erode power for analysis. Specifically, dividing into more refined categories will lead to lower sample size per category and an increase in the number of parameters to be estimated. The reviewer might have noted that we already could not include bidirectional sex changers and simultaneous hermaphrodites in some of our analyses precisely because of their low sample sizes. However, we agree with the reviewer that the way we have dismissed diandric and digynic species does not do justice to the incredible and complex variation in sexual systems. Thus, in response to the reviewer’s comment we have now discussed this (Lines 231-235 and lines 549-550).

d. Table 1 – mating system definitions are confusing and even misleading; (i) gonochoristic species can have harems (ii) what does the double arrow mean in column two, for gonochore species or protogynous species, for example? (ii) should specify if ‘sex ratio’ is adult sex ratio (which presumably it is) (iv) I don’t really understand the male/female symbols in the last column. For example, if species is biased to one or the other sex why are the sexes equal in number?

Thank you for checking with attention Table 1. Relating to your general comment, please see also our response under 1.a outlined above. Regarding the specific questions, we note that Table 1 already reflects that gonochoristic species can have harems (the last example in column 2 under gonochorism); the double arrow indicates the sexual interactions between the two sexes (Table 1 legend for mating systems: monogamy (pair bond) or random pairing: 1 male interacting with 1 female ($\text{♂} \leftrightarrow \text{♀}$); promiscuity: multiple males interacting with multiple females: $\text{♂♂♂} \leftrightarrow \text{♀♀♀}$). We also address the reviewer’s comment and now clarify

that sex ratio is adult sex ratio and, as we now indicate in the footnotes of Table 1, symbols do not reflect bias in sex ratios but a broad classification of spawning behavior (from pair spawning to group spawning/spawning aggregations). Please see also our response to Reviewer 1, who raised a related point, where we clarify that Table 1 summarizes the main predictions derived from theory and applicable to most common sexual systems; it is not meant to identify predictions for the most peculiar and rarest cases.

2. Database:

a. Many of the 4741 species listed with their life history characters in the SI have an assumed sexual pattern indicated (from FishBase?) which has not been confirmed. Note that FishBase sometimes makes generic assumptions on certain life history features, such as sexual pattern, so these need to be checked species by species in the literature. A wealth of life history parameters relevant to this paper can be found in publications of the last few decades.

We totally agree with the reviewer and indeed this is exactly what we did; we apologize for not making this clear in our first submission. Specifically, in our original submission we used FishBase to get a first list of species for which there is information on sexual system and then searched the primary literature for each individual species to verify—and correct if necessary—the assignment to sexual system (date of initial search: June 2018; last search: June 2021). For some species we could not find any reference confirming the sexual system as indicated in FishBase.

In response to the concerns of the reviewer, therefore, we have taken further steps to ensure the thoroughness of our dataset and the robustness of our results:

- 1) we have repeated a literature search to confirm the classification of sexual system of the species in our dataset (date of last search: June 2021).
- 2) We now include in the analyses only hermaphroditic species for which we can confirm the sexual system (294 in the current submission; See Supplementary Information).

As mentioned at the beginning of our response, most of our conclusions remain unaltered with the exception of the direct transition from gonochorism to simultaneous hermaphrodites, which we now find it is unlikely, and the larger size of protandrous species, which is now no longer supported.

b. There appear to be only 271 species in the SI for which mating system (gonochorism versus hermaphroditism type) is actually confirmed and some of these are incorrect and outdated. For example, published literature after 2000 for *Epinephelus polyphekadion* gives the needed life history parameters and shows gonochorism and not protogyny for this species. A full comprehensive updated literature review is needed to prepare the database for analysis.

We thank the reviewer for this suggestion and thorough check of our dataset; this and the above comment has led us to check once more our data and performed a new search for primary sources (see above). For *E. polyphekadion* Mapleston et al. (2009) find that his species has transitional individuals, while Rhodes et al. (2011) state that the species exhibits “functional gonochorism with the potential for protogynous sexual transition”. In this

specific case, therefore, we have removed the species from our dataset given the uncertainty still present on its sexual system.

3. Predictions:

a. The basic premise of the paper is summarized as: ‘Since sequential hermaphrodites achieve higher fitness when reproducing as the second sex, they should benefit more from increased longevity or larger size than gonochoristic and simultaneous hermaphroditic species. Alternatively, sequential hermaphrodites could mature earlier as the first sex compared to the same sex in gonochoristic species and capitalize on reproduction as second sex. These predictions, however, remain to be tested.’ I do not understand the basic premise here. It needs to be explained more clearly (not just by referring to another paper). For example, the selective pressure for sexual pattern is within species and not across species i.e. that under certain environmental circumstances or mating systems large size confers greater RS in males in protogynous species within that species than gonochorism. Features such as longevity or maximum size are constrained or determined by other factors within species. Moreover, for many fishes maximum length and size of maturation are correlated and these can vary according to environmental conditions (e.g. within a species, in colder environments, growth is overall slower and that affects max size and maturation size).

We totally agree with the reviewer that selection acts on populations within species but, when selection varies across species, it ultimately produces interspecific differences that are expected according to theory. This is the underpinning principle of large-scale comparative studies like ours (Harvey & Pagel, 1990). Theoretical models for the evolution of sequential hermaphroditism, based on selection at the population/species level, suggest that for sex changers, higher fitness is achieved as second sex (formally tested for 8 species by Benvenuto et al., 2017) - this explains the advantage of changing sex. We can then bring this into the context of life history theory. Specifically, if the second sex has higher fitness, sequential hermaphrodites should benefit from living longer (overall and/or as a second sex in particular, and/or by maturing as second sex earlier), being larger (especially in protandry where females are the larger sex and size gives fecundity advantage). We agree that, as most species traits, life history traits (and sexual system) are under several selective forces; however, theory suggests that life history traits should differ predictably between sexual systems as stated above. This has now been detailed in the introduction (Lines 163-168). In our study we aim to investigate the predictions derived from theory and establish generality of principles across a large number of species.

We also agree with the reviewer that maximum length and size at maturity covary and respond to environmental selective pressures; indeed, this would be a great question for future studies when more sex-specific life history trait become available. To address these and also similar comments by Reviewer 1, we have now repeated our analyses correcting for allometry (including correction for age at maturity with size at maturity, as we have sex-specific data on this variable and not total length (Lines 329-337)).

4. Phylogeny:

a. Phylogenetic relationships within the teleosts are poorly understood. It is not clear what system of phylogeny is applied in this paper. The most recent phylogeny for Actinopterygii

fishes that I am aware of, not cited in this paper, is by Hughes et al., 2018 PNAS vol 115. Comprehensive phylogeny of ray-finned fishes...., available online.

Here we have used the molecular phylogeny by Rabosky et al. (2018; *Nature*) which includes 11638 species in total. We had extracted data on sexual system and life history traits of 9005 species. Of these, 4614 were present in both the Phylogeny of Rabosky and in our revised dataset. However, the paper with the genomic tree by Hughes et al. includes only 303 species *in total*, well below the number of species in our dataset (over 4600 species), and thus it is not suitable for the purpose of our study.

Reviewer #3 (Remarks to the Author):

This manuscript presents a very nice comparative analysis of sexual systems in teleosts. The scope of the analysis far exceeds earlier studies and allows the authors to gain significant new insights into the controls of vertebrate sexual systems. I think these large scale evolutionary studies can be important for advancing a field because they provide the opportunity to examine widespread ideas that have been gained from case studies on smaller sets of species. However, the current manuscript does not hit it out of the park for me in communicating the most significant findings. There are some really good ideas in the tables and figures but they are not yet polished to deliver a cohesive story. For example, fig 2 desperately needs a legend and more polish in general. Why are some lineages labelled? Why are some lineages represented with icons and not others? Why not colour code the branches to indicate some information about the analysis?

There should be clear visual representation of the characters under analysis here. You might be able to incorporate rate information here as well. I liked figure 1 but then wondered why the colour and icons here were not carried through to figs 3 and elsewhere. The problem here is that there are four character states and the possible transitions between them are complex. The hypothesis space is also complex so the authors really need to do a better job of bringing clarity to the analyses. I think color coded icons would work much better in fig 3 than letter abbreviations. The rate comparison doesn't really come through in table 2. I would try plotting multiple posteriors on the same graph--the table might go to the SI. Fig 4 should be boxplots rather than bar plots. Similarly, I wonder if there is a way to visually represent the pgl results as a multipanel scatter plot with the significant relationships indicated. The full table might go to the SI.

We thank the reviewer for considering our study comprehensive and for providing new insights on vertebrate sexual systems and thank for these useful suggestions that have improved our study. We agree that Figure 2 was too complex and have followed the reviewer's suggestion to improve it as follows:

1. We now present the distribution of sexual system on the phylogeny as 5-character states (gonochorism, simultaneous hermaphroditism, protandry, protogyny, bidirectional sex change) as a separate figure (Figure 2) using the same colour coding that will be subsequently used for the results at 4-character states (bidirectional sex changers are excluded for this analysis due to small sample size). Further, in this

figure we have added the missing silhouettes for all families which include hermaphroditic species;

2. We present the results of the evolutionary history (ancestral state reconstruction and transition rates) as a separate, multipanel figure (Figure 3) depicting: (i) the ancestral state reconstruction across the tree using a stronger colour (magenta) to colour code hermaphroditism; (ii) retain the posterior distribution of the character state at the root; (iii) add, as per suggestion, the posterior distribution of the transition rates.
3. We have moved the table of the estimated transition rates for the analysis at 2 states into the SI, as suggested.
4. We retain the colour code of sexual system also for Table 1 (predictions) to ensure consistency across the whole ms.
5. Following Reviewer 1's suggestion, names of sexual system are now spelled out in the figure of results of the transition rates among 4 sexual systems (now Figure 4, previously Figure 3 in the earlier submission). In this figure we have also added the requested icons (now added also in Table 1).
6. Regarding the panels of former figure 4 (now Figure 5), these do represent the output of the PGLS models (phylogenetic means and phylogenetic SE for analyses without allometry; phylogenetic fit lines for traits analyzed with allometric control). For these reasons we use bar plots that depict means and SE, hence matching exactly the PGLS output, rather than boxplots that depict non-phylogenetic medians and interquartile ranges.

I found two issues with the discussion. First the ideas of Pennell et al seem highly relevant to the paper and should be incorporated into the introduction as part of hypotheses to test. Right now this discussion of refuting their study comes as a surprise. The discussion spends a lot of space on it so I think you would be better served by incorporating this earlier.

Thanks for this suggestion: we have included the reference to Pennell et al in the introduction on the importance in studying transition rates in teleosts (line 242) and mentioned them as an example of the common current approach to analyze hermaphroditism and gonochorism as binary traits (Line 125). We agree with Pennell et al's study regarding the reversibility of the transition from hermaphroditism to gonochorism (which we now explicitly acknowledged in the text (Line 389), but we have found opposite magnitude of the transition rates between gonochorism and hermaphroditism. In our discussion we thus comment on the differences between our dataset and Pennell et al's, which we think underpins this difference in the results (Lines 394-399).

Second, I found the discussion to be somewhat long and unfocused. It feels like the authors are trying to say everything here. I did not find the presentation of a generalized model for studying sexual systems to be especially helpful and I wonder if it might be better to show in a review paper rather than this study. Subheadings might be useful as well.

We feel that the presentation of a general model for studying sexual systems should follow the discussion of our results on the evolutionary history of sexual systems in teleosts. Our study has evidenced some knowledge gaps and we are convinced that progress in this field requires a new framework. However, we acknowledge that maybe this was not clear enough in the original version. Thus, in the revised manuscript not only we mention that we present

a general model for studying sexual systems (Lines 460-464) but also we have tried to improve the flow and clarity of the discussion, adding the two main points, i.e., sex determining mechanisms (starting at Line 472) and gonadal plasticity (starting at Line 493), individually.

Overall I think the study has potential but I don't feel like it quite hits the mark in its current version.

Some final small comments. I think the authors are too colloquial with the designations of fish lineages. It is fine to be informal with the taxonomic designations after you have defined the major lineages but right now you talk about fish in an informal way at the start of the paper before injecting systematic terms. I think you need a short description of the taxonomic focus of the study and a justification of them.

Thank you for pointing this out. Throughout the revised manuscript we now are more specific and consistently use the word 'teleosts'.

I found it odd that the source phylogeny is not cited in 1205. The tree files for that paper are distributed via a website so if the authors used that it would be also appropriate to cite J. Chang, D. L. Rabosky, S. A. Smith, An R package and online resource for macroevolutionary studies using the ray-finned fish tree of life. *Methods Ecol. Evol.*

The source phylogeny was already cited in the Methods section in the original submission. However, following this reviewer request, in the revised version it is now explicitly cited also at the end of the introduction (Line 227). Please note that we have not used the R package but the dryad data from the original publication (<https://doi.org/10.5061/dryad.fc71cp4>). We have also added a link to The Fish Tree of Life website to make it easier for the readers to find it (Line 545).

Fig 2 caption. Citation to ape is not needed. A legend explaining color key and internal external circles is.

Figure 2 has now been changed into two separate figures and this should help reading it. In the legend the citation to ape has been removed. Please see our response earlier about changes to Figure 2 (and former Figure 3).

Reviewers' Comments:

Reviewer #1:

Remarks to the Author:

I was the Reviewer 1 of the former version, I like the manuscript a lot already in the original version; however, at the same time, I had several critical comments. I feel that all of them were very carefully considered and incorporated into the manuscript. The new version is much improved, much easier to read and much more focused. The new Figure 3 is very illustrative. To me the study is very important and I believe that it will be read with a great interest by a wide audience.

I have just a single personal recommendation to the authors (that might be ignored): the paper is not directly about mating systems, you analysed only sexual systems (gonochorism, protogyny etc.) and life-history data. I agree with the Reviewer 2 that mating systems of teleosts are very variable and not so easy to be simply classified. Therefore, I feel that especially Table 1 can attract criticism. I understand that a broad study as this one has to make some generalizations, but still I would soften the statements in Table 1 even more. For example, why gonochoristic species should have adult sex ratios 1:1? Gonochoristic species can have environmental sex determination (although I am aware that ESD is rare in teleosts from the earlier work of a member of the authors' team) and ESD does not ensure balanced sex ratio at hatching. More importantly, sex-biased mortality is common, and we can expect for example male-biased mortality in teleosts with well-developed male ornaments as demonstrated in guppies and killifish. And the diversity of mating systems and adults sex ratios in gonochoristic lineages is indeed wide (e.g. in mammals and birds). I think that mating systems are important here to take a broader perspective on consequences of sexual system and to derive hypotheses, but I feel that still strong expressions as presented in the Table 1 can lead to unnecessary criticism. At the same time this is not so crucial point for the study. I would even more reduce the parts about mating systems and move them more to discussion.

Reviewer #2:

Remarks to the Author:

The manuscript has two key components, the phylogenetic analysis of the incidence of different forms of sexual patterns and the possible relationships/adaptive significance of these according to selected life history traits, in particular size/age (max and sex mat), and male GSI, longevity. This is an interesting analysis the major strength of which is the phylogenetic component.

The authors are clearly stronger on the phylogenetic than the life history analysis component as highlighted below. As such, the life history component still has weaknesses, most of which can be easily addressed in particular in relation to terminology, but, overall, does not really provide much in the way of new insights. This is largely due to the lack of the necessary data (in the literature) on life history attributes and mating systems. Detailed comments on the revised document are provided below:

ABSTRACT : the abstract focus is on phylogeny/evolution and not much on the mating systems/adaptive components that the paper cover.

TERMINOLOGY: it is more common in the literature to refer to 'sexual patterns' for type of reproductive mode (i.e. gonochorism, hermaphroditism) rather than 'sexual system'. Sexual patterns is a term that also helps to distinguish these from 'mating systems'.

STRUCTURE: I suggest that each manuscript section (e.g. Intro, Results..) be divided into the appropriate section i.e. phylogeny or life history, e.g. one example is around line 133 where a subtitle would be helpful.

METHODOLOGY: The GSI is defined as the percentage of body mass devoted to the gonads and is

used as an indicator of intensity of sperm competition. It is not stated specifically how the GSI data were compiled. For example, to be a reliable indicator of sperm competition the GSI value used would have to be measured at its peak in the reproductive season. To use male GSI outside of the spawning season would be meaningless but there is no indication that maximum GSI during the spawning season was the value used for GSI in the analysis. This should be clarified in methods.

MATING SYSTEM/SPAWNING BEHAVIOR CATEGORIES: The following, as raised in the original reviewer comments, needs attention/correction (Lines 182-184, 192-195, Fig. 6 and elsewhere). It is not true that spawning behaviour is broadly classified in fish as pair spawning, involving only two individuals at the time, and spawning aggregations, comprising large breeding groups'. The divisions are pair/group spawning and monogamy/polygamy.

The authors are confusing mating system and spawning behaviour (and this confusion is easy to solve!!). For example in Fig. 6 - MATING SYSTEM is divided into monogamy/random pairs, and polygyny/group mating and SPAWNING BEHAVIOUR is divided into pair-spawning and spawning aggregations. As raised in the original review, pair and group spawning are very different and, indeed, this is important to highlight for this paper because they tend to be associated with the male GSI differences i.e. pair-spawners with small max male GSI and group-spawners with high max male GSI. This is all in line with the GSI analysis in this paper which works fine. However, the authors confuse the issue by unnecessarily associating 'group' spawning with aggregations specifically. Aggregations can have either group or pair-spawning. This is NOT a peculiarity, unusual aspect or oddity of a mating system and occurs in a great many species, so cannot be dismissed as the authors suggest. Quite simply, aggregations and group spawning are NOT synonymous.

Actually, the analysis of GSI is a good one in this paper and reflects pair and group-spawning well. It is not even necessary to mention aggregation spawning (WHICH IS NOT THE SAME AS GROUP SPAWNING)! So, for Fig. 6 the SPAWNING BEHAVIOUR could simply be divided into pair-spawning and group-spawning (remove spawning aggregations entirely). For MATING SYSTEM just use monogamy or polygamy.

Why is the GSI component not included in Table 1 – that could be helpful.

SEXUAL PATTERN (sexual system) categories: The authors state (numbers are lines): "We compiled the most comprehensive database 550 on sexual systems in teleosts to date. Information on sexual system were first extracted from FishBase94551 . Next, species were 552 classed as hermaphroditic only if functional hermaphroditism could be confirmed by primary literature, as recently compiled elsewhere95 553 (see Supplementary Data for details).".

I understand from the review conducted that about 100 species of hermaphrodite were identified following review of the literature. There are many more gonochore species. However, I cannot see how gonochorism was determined/confirmed according to the methods provided. To my knowledge there are not so many studies that definitively confirm gonochorism. Please could the authors clarify whether they assumed gonochorism in all species that were not clearly determined to be hermaphroditic, or, alternatively, only applied gonochorism to species for which the literature has definitely confirmed gonochorism? If the latter, then this literature is not identified in the SI.

FOR REVISION: Lines 180-188 I disagree that the assumption of larger absolute size/age associated with sexual pattern at the species level is valid. This is not supported by theory (such as size advantage model) since the importance of size in sex changing species is not an absolute one but one of relative size i.e. the largest compared to others.

LENGTH: The text is somewhat long and a little repetitive for the phylogeny sections. Can these be more concise and reduce repetition?

RESPONSE TO REVIEWERS

Reviewer #1:

I was the Reviewer 1 of the former version, I like the manuscript a lot already in the original version; however, at the same time, I had several critical comments. I feel that all of them were very carefully considered and incorporated into the manuscript. The new version is much improved, much easier to read and much more focused. The new Figure 3 is very illustrative. To me the study is very important, and I believe that it will be read with a great interest by a wide audience.

We thank the reviewer for his/her very positive appraisal of our work and the constructive criticism.

I have just a single personal recommendation to the authors (that might be ignored): the paper is not directly about mating systems, you analysed only sexual systems (gonochorism, protogyny etc.) and life-history data. I agree with the Reviewer 2 that mating systems of teleosts are very variable and not so easy to be simply classified. Therefore, I feel that especially Table 1 can attract criticism. I understand that a broad study as this one has to make some generalizations, but still I would soften the statements in Table 1 even more. For example, why gonochoristic species should have adult sex ratios 1:1? Gonochoristic species can have environmental sex determination (although I am aware that ESD is rare in teleosts from the earlier work of a member of the authors' team) and ESD does not ensure balanced sex ratio at hatching. More importantly, sex-biased mortality is common, and we can expect for example male-biased mortality in teleosts with well-developed male ornaments as demonstrated in guppies and killifish. And the diversity of mating systems and adults sex ratios in gonochoristic lineages is indeed wide (e.g. in mammals and birds).

The reviewer acknowledges that a broad study as this one has to make some generalizations but he/she has a good point on adult sex ratio variability in gonochoristic species. Indeed, we realised the main point we wanted to make was the more consistent sex ratio skew towards the first sex in sex changers. Following the reviewer's recommendation, we have thus changed the adult sex ratio to "variable" for gonochoristic species. In addition, we moved our previous disclaimer about the generality of predictions and possible exceptions from the bottom of the table to the main legend to incorporate the reviewer's point. We thank the reviewer for this suggestion.

I think that mating systems are important here to take a broader perspective on consequences of sexual system and to derive hypotheses, but I feel that still strong expressions as presented in the Table 1 can lead to unnecessary criticism. At the same time this is not so crucial point for the study. I would even more reduce the parts about mating systems and move them more to discussion.

The review has correctly understood that we are interested in mating systems. Although we could not analyse them formally, given the lack of sufficient data, we felt the need to mention how they may influence the direction of sex change, as recognised by the size-advantage model (line 147-158). We thus included them in the introduction alongside spawning behaviour that relates to GSI. We recognize Table 1 and the Introduction provide broad generalizations (given the high variability in fish strategies), but we feel that paragraph 147-158 is necessary as it covers the

essential background on hermaphroditism for readers not familiar with the topic and decided to keep it.

We thank both reviewers for their help to make table 1 more accurate.

Reviewer #2:

The manuscript has two key components, the phylogenetic analysis of the incidence of different forms of sexual patterns and the possible relationships/adaptive significance of these according to selected life history traits, in particular size/age (max and sex mat), and male GSI, longevity. This is an interesting analysis the major strength of which is the phylogenetic component.

The authors are clearly stronger on the phylogenetic than the life history analysis component as highlighted below. As such, the life history component still has weaknesses, most of which can be easily addressed in particular in relation to terminology, but, overall, does not really provide much in the way of new insights. This is largely due to the lack of the necessary data (in the literature) on life history attributes and mating systems. Detailed comments on the revised document are provided below:

We thank the reviewer for the detailed revision of our paper and the constructive criticism. We note that we emphasize in the discussion that there is urgent need for more data on life history traits to evaluate more in depth the role of life histories in the evolution of sexual systems.

ABSTRACT : the abstract focus is on phylogeny/evolution and not much on the mating systems/adaptive components that the paper cover.

Following the reviewer suggestion, we have briefly (given the word limit) added the importance to invest more in the future on mating/spawning behaviours (lines 44-45).

TERMINOLOGY: it is more common in the literature to refer to 'sexual patterns' for type of reproductive mode (i.e. gonochorism, hermaphroditism) rather than 'sexual system'. Sexual patterns is a term that also helps to distinguish these from 'mating systems'.

In the literature both "sexual system" and "sexual pattern" are used. We have followed the definition of Dr Janet Leonard (as also used in the title of her 2019 seminal book: "The evolution of sexual systems in animals"; Springer). Furthermore, in our study on sexual systems in fish we discuss our findings in a broad context and refer to the situation in other taxa including crustaceans and plants, where the term "sexual system" is mostly used. Finally, in our previous papers (refs. #58 & #96) we also used this term. Therefore, we would prefer to keep the use of "sexual system". However, we do recognize that "sexual patterns" is also commonly used, thus we have added it in the initial definition: "Sexual systems (also known as sexual patterns), defined as the pattern of distribution of the male and female function among the individuals of a given species..." [the underlined words have been added for clarity] (lines 55-57).

STRUCTURE: I suggest that each manuscript section (e.g. Intro, Results.) be divided into the appropriate section i.e. phylogeny or life history, e.g. one example is around line 133 where a subtitle would be helpful.

We have observed that papers published in Nature Communications have commonly subsections in Results and Material and methods, but not in the Introduction and Discussion, and we have

structured our manuscript accordingly. We will follow the reviewer's suggestion if helpful to the readers and recommended by the Editor given the journal's style .

METHODOLOGY: The GSI is defined as the percentage of body mass devoted to the gonads and is used as an indicator of intensity of sperm competition. It is not stated specifically how the GSI data were compiled. For example, to be a reliable indicator of sperm competition the GSI value used would have to be measured at its peak in the reproductive season. To use male GSI outside of the spawning season would be meaningless but there is no indication that maximum GSI during the spawning season was the value used for GSI in the analysis. This should be clarified in methods.

This is a good point. We indeed used the male maximum GSI, which is expected to coincide with the peak of the species reproductive season. We have now added this information in the manuscript (lines 558-559) and in the supplementary information. Thanks for pointing it out to us.

MATING SYSTEM/SPAWNING BEHAVIOR CATEGORIES: The following, as raised in the original reviewer comments, needs attention/correction (Lines 182-184, 192-195, Fig. 6 and elsewhere). It is not true that spawning behaviour is broadly classified in fish as pair spawning, involving only two individuals at the time, and spawning aggregations, comprising large breeding groups'. The divisions are pair/group spawning and monogamy/polygamy.

The authors are confusing mating system and spawning behaviour (and this confusion is easy to solve!!). For example in Fig. 6 - MATING SYSTEM is divided into monogamy/random pairs, and polygyny/group mating and SPAWNING BEHAVIOUR is divided into pair-spawning and spawning aggregations. As raised in the original review, pair and group spawning are very different and, indeed, this is important to highlight for this paper because they tend to be associated with the male GSI differences i.e. pair-spawners with small max male GSI and group-spawners with high max male GSI. This is all in line with the GSI analysis in this paper which works fine. However, the authors confuse the issue by unnecessarily associating 'group' spawning with aggregations specifically. Aggregations can have either group or pair-spawning. This is NOT a peculiarity, unusual aspect or oddity of a mating system and occurs in a great many species, so cannot be dismissed as the authors suggest. Quite simply, aggregations and group spawning are NOT synonymous.

Actually, the analysis of GSI is a good one in this paper and reflects pair and group-spawning well. It is not even necessary to mention aggregation spawning (WHICH IS NOT THE SAME AS GROUP SPAWNING)! So, for Fig. 6 the SPAWNING BEHAVIOUR could simply be divided into pair-spawning and group-spawning (remove spawning aggregations entirely). For MATING SYSTEM just use monogamy or polygamy.

We thank the reviewer for pointing this out. In the revised version, we classify spawning behaviour as pair-spawning and group spawning (lines 181-183 and 188-193). We have also modified Fig 6 accordingly.

Why is the GSI component not included in Table 1 – that could be helpful.

GSI can be quite difficult to categorise, as mating systems and spawning behaviours combined determine the intensity of sperm competition. Following the reviewer request, we have tried to add high GSI or low GSI for each mating system (in each sexual system), but the final table becomes difficult to read. Thus, we would prefer not to add GSI in Table 1.

SEXUAL PATTERN (sexual system) categories: The authors state (numbers are lines): “We compiled the most comprehensive database 550 on sexual systems in teleosts to date. Information on sexual system were first extracted from FishBase94551 . Next, species were 552 classed as hermaphroditic only if functional hermaphroditism could be confirmed by primary literature, as recently compiled elsewhere95 553 (see Supplementary Data for details).”.

I understand from the review conducted that about 100 species of hermaphrodite were identified following review of the literature. There are many more gonochore species. However, I cannot see how gonochorism was determined/confirmed according to the methods provided. To my knowledge there are not so many studies that definitively confirm gonochorism. Please could the authors clarify whether they assumed gonochorism in all species that were not clearly determined to be hermaphroditic, or, alternatively, only applied gonochorism to species for which the literature has definitely confirmed gonochorism? If the latter, then this literature is not identified in the SI.

We thank the reviewer for pointing this out. Our dataset included 4320 gonochoristic and 294 hermaphrodite species. Species were classed as hermaphroditic only if functional hermaphroditism could be confirmed by primary literature. As the reviewer rightly notes, gonochorism is rarely determined even when present in fish. Therefore, considering gonochoristic only species for which this sexual pattern is confirmed would strongly bias the dataset against gonochorism, ultimately undermining the robustness of the analyses. Thus, following the reviewer’s suggestion, we considered gonochoristic only species regarded as such in FishBase, unless recent literature states otherwise. Importantly, species for which there is contrasting information were discarded and not used for this study (Supplementary figure 1). We state this in the Introduction (new sentences; lines 231-234) to make it clear early to readers, but also in the Materials and Methods (lines 543-549), as well as in the Supplementary information (last two sentences in the “Data collection and verification” section; page 6).

FOR REVISION: Lines 180-188 I disagree that the assumption of larger absolute size/age associated with sexual pattern at the species level is valid. This is not supported by theory (such as size advantage model) since the importance of size in sex changing species is not an absolute one but one of relative size i.e. the largest compared to others.

We could not find in the text any reference to size in lines 180-188. The reviewer might refer to lines 170-177 of the previous submission, where we discussed broadly different strategies across species with different mating systems. Thus, while it is true that relative size is important at the intra-specific level for individual to “decide” to change sex, in lines 173-174 we discussed the benefit of larger size in females. In general, larger females produce more eggs than smaller ones both within and across species, while larger males do not necessarily increase sperm production with size. In males, larger size gives an advantage to secure dominance/change sex (and increase fertilization rates), but not fecundity. To address the reviewer’s point, we have added some additional explanation in the text (lines 174-177) .

LENGTH: The text is somewhat long and a little repetitive for the phylogeny sections. Can these be more concise and reduce repetition?

We have re-read the text carefully and we have removed a couple of sentences at the beginning of the discussion. We believe that some limited repetition of the results of the phylogenetic analysis is unavoidable, but every time we have been unlocking a different aspect. We summarized the

results at the end of the introduction as required by the journal; presented them in the results section and then discussed them in detail to address specific angles: the discrepancy with a previous paper; the relevance in the theoretical framework of sequential hermaphroditism as an intermediate step and fact that androdioecy cannot serve that role. We felt that these are complex topics and require space to be fully discussed. For this reason, we would prefer to leave the text in its current form, unless the reviewer feels that this is a major aspect to be resolved.

Reviewers' Comments:

Reviewer #2:

Remarks to the Author:

This is an interesting paper that is strong on the phylogeny side (with one exception) see below and weaker on the mating system side. The following comments have both been included in previous reviews and are not yet fully addressed; the first one does need to be fully addressed. It is further suggested that the authors consider removing Table 1 as being confusing and incomplete. There are two remaining items to address, the first being particularly important.

Gonochorism and Hermaphroditism Assignment

...The authors say: 'Thus, following the reviewer's suggestion, we considered gonochoristic only species regarded as such in FishBase, unless recent literature states otherwise....We state this in the Introduction (new sentences; lines 231-234) to make it clear early to readers.....'

Lines 231- 234 still do not address any concern about overestimating gonochorism, that I can see, or make the above caveat regarding gonochorism as a default sexual pattern. Here is the current text: 'Our large-scale approach allows us to fully unravel how sexual patterns evolved 232 and identify which ones represent evolutionary stable conditions. We focus on gonochorism, 233 protogyny, protandry and simultaneous hermaphroditism as these are the most common sexual 234 system in teleosts.' There is no mention here about 'overestimating' gonochorism.

However, in the Materials and Methods, the authors do state: 'We compiled the most comprehensive database on sexual systems in teleosts to date. Information on sexual system were first extracted from FishBase. Next, species were classed as hermaphroditic only if functional hermaphroditism could be confirmed by primary literature, as recently compiled elsewhere (see Supplementary Data for details). We classified as gonochoristic all species that were not clearly determined to be hermaphroditic.'

This last line is an appropriate edit and the approach is acceptable. However, authors still need to address in the Discussion how using gonochorism as the default sexual pattern in the absence of evidence for hermaphroditism, could affect their analysis and interpretation therefrom.

The authors note in a response on this issue: 'As the reviewer rightly notes, gonochorism is rarely determined even when present in fish. Therefore, considering gonochoristic only species for which this sexual pattern is confirmed would strongly bias the dataset against gonochorism, ultimately undermining the robustness of the analyses'. I

I agree with the authors here but this situation now means that there is a bias TOWARDS gonochorism which could influence the outcomes of their analyses.

The reason, as I stated before, is that Fishbase and many of the citations its uses for sexual patterns, such as Breder and Rosen and FAO species catalogs, have not actually determined sexual patterns. So the 'gonochore' species in the current publication are, in fact, mostly species that are 'not diagnosed as hermaphroditic'. This merits discussion in the context of the outcomes of the analysis and possible implications.

As just one of many possible examples, *Lates nilotica*, the Nile perch, is listed as a gonochore although several papers say that it is likely to be hermaphroditic because of male/female size distributions and other factors. It is also related to the protandrous congener *L. calcarifer* (a proven hermaphrodite). So hermaphroditism is certainly very likely but is not confirmed and, hence, the species is listed in the default gonochore category.

Since designation of 'gonochore' for non-hermaphrodites is a major and inevitable flaw of the paper, it needs to be clearly discussed for its implications.

Mating system

Table 1 – I find Table 1 to be confusing and not helpful. I would suggest that the authors consider removing it because it further weakens the mating system part of this paper, in my opinion. For example spawning behaviour options (last column) do not include group-spawning of single females with multiple males, which is an important type of group-spawning associated with large male GSIs in both gonochore species and in primary males of hermaphroditic (diandric) species.

I still feel that the authors do not understand that 'group-spawning' involves either multiple males and females (and hence males competing spermwise) or, and differently, a female and multiple males (again with sperm competition). These are the two types of 'group-spawning' in both hermaphroditic and gonochore species. Such information appears in many publications and is not exceptional or rare.

Reviewer #3:

Remarks to the Author:

Dear Editor,

I found the revised version of this paper to be substantially improved and am now convinced that this is an important study that provides a strong basis for future investigations of the controls of sexual system evolution in fishes and other organisms. My only substantial comment is that the authors conclude that hermaphroditism does not evolve directly from gonochorism. I agree that the low transition rates support this conclusion but the authors could also directly test this within BayesTraits by comparing models where the rates are set to 0 or not allowed to reversible models. I think this would be desirable given the importance of the result to their paper. On a minor note there is a formatting error in figure 1 in the first row where the symbols for the predicted mating systems are on the same line instead of on separate lines.

Response to reviewers of NCOMMS-21-02788B **“Switches, stability and reversals in the evolutionary history of sexual systems in fish”**.

Reviewer #2

This is an interesting paper that is strong on the phylogeny side (with one exception) see below and weaker on the mating system side. The following comments have both been included in previous reviews and are not yet fully addressed; the first one does need to be fully addressed. It is further suggested that the authors consider removing Table 1 as being confusing and incomplete. There are two remaining items to address, the first being particularly important.

Gonochorism and Hermaphroditism Assignment

...The authors say: 'Thus, following the reviewer's suggestion, we considered gonochoristic only species regarded as such in FishBase, unless recent literature states otherwise.....We state this in the Introduction (new sentences; lines 231-234) to make it clear early to readers.....'

Lines 231- 234 still do not address any concern about overestimating gonochorism, that I can see, or make the above caveat regarding gonochorism as a default sexual pattern. Here is the current text: 'Our large-scale approach allows us to fully unravel how sexual patterns evolved 232 and identify which ones represent evolutionary stable conditions. We focus on gonochorism, 233 protogyny, protandry and simultaneous hermaphroditism as these are the most common sexual 234 system in teleosts.' There is no mention here about 'overestimating' gonochorism.

However, in the Materials and Methods, the authors do state: 'We compiled the most comprehensive database on sexual systems in teleosts to date. Information on sexual system were first extracted from FishBase. Next, species were classed as hermaphroditic only if functional hermaphroditism could be confirmed by primary literature, as recently compiled elsewhere (see Supplementary Data for details). We classified as gonochoristic all species that were not clearly determined to be hermaphroditic.'

This last line is an appropriate edit and the approach is acceptable. However, authors still need to address in the Discussion how using gonochorism as the default sexual pattern in the absence of evidence for hermaphroditism, could affect their analysis and interpretation therefrom.

The authors note in a response on this issue: 'As the reviewer rightly notes, gonochorism is rarely determined even when present in fish. Therefore, considering gonochoristic only species for which this sexual pattern is confirmed would strongly bias the dataset against gonochorism, ultimately undermining the robustness of the analyses'.

I agree with the authors here but this situation now means that there is a bias TOWARDS gonochorism which could influence the outcomes of their analyses.

The reason, as I stated before, is that Fishbase and many of the citations its uses for sexual patterns, such as Breder and Rosen and FAO species catalogs, have not actually determined sexual

patterns. So the 'gonochore' species in the current publication are, in fact, mostly species that are 'not diagnosed as hermaphroditic'. This merits discussion in the context of the outcomes of the analysis and possible implications.

As just one of many possible examples, *Lates nilotica*, the Nile perch, is listed as a gonochore although several papers say that it is likely to be hermaphroditic because of male/female size distributions and other factors. It is also related to the protandrous congener *L. calcarifer* (a proven hermaphrodite). So hermaphroditism is certainly very likely but is not confirmed and, hence, the species is listed in the default gonochore category.

Since designation of 'gonochore' for non-hermaphrodites is a major and inevitable flaw of the paper, it needs to be clearly discussed for its implications.

We are sorry for the confusion and thank the reviewer for his/her constructive criticism. In the revised version we start early in the Introduction by acknowledging that non-hermaphroditic species are classified as gonochoristic (lines 204-207). Furthermore, in the Discussion we have devoted an entire new paragraph to deal with this issue. This paragraph (lines 311-323) includes the following explanation:

"We have accepted the classification in FishBase⁶² for gonochoristic species (unless rejected or disputed by primary literature), without individually confirming their sexual system as done for the hermaphroditic species in our dataset. This is because gonochorism is rarely confirmed in primary source even when present in fish. As a result, if we used only the few gonochoristic species for which sexual system is explicitly confirmed in the original sources, the dataset would be strongly biased against gonochorism and include an unrealistic small number of gonochoristic species, ultimately undermining the robustness of the results. However, we acknowledge that a few species currently classified as gonochoristic in our dataset might be hermaphroditic. Although it is not possible to predict how this could influence the outcome of the analysis, given that this depends on the number of affected species, their phylogenetic position and the sexual system of their closely related species, our results represent an accurate picture of the evolution of sexual system in fish with the data currently available"

Finally, in the Methods section (lines 456-463) we have the following text:

"Next, species were classed as hermaphroditic only if functional hermaphroditism could be confirmed by primary literature, as recently compiled elsewhere⁹⁶, with further species confirmed from the primary literature. For the remaining species, we maintained the gonochoristic classification of FishBase⁶², unless recent literature stated otherwise. Indeed, gonochorism is rarely confirmed in literature even when present, so including as gonochoristic only species for which this sexual pattern is confirmed would strongly bias the dataset against gonochorism, ultimately undermining the robustness of the analyses"

We hope that this will satisfactorily address the concerns about classifying as gonochoristic the species for which functional hermaphroditism has not been confirmed.

Mating system

Table 1 – I find Table 1 to be confusing and not helpful. I would suggest that the authors consider removing it because it further weakens the mating system part of this paper, in my opinion. For example spawning behaviour options (last column) do not include group-spawning of single females with multiple males, which is an important type of group-spawning associated with large male GSIs in both gonochore species and in primary males of hermaphroditic (diandric) species.

I still feel that the authors do not understand that ‘group-spawning’ involves either multiple males and females (and hence males competing spermwise) or, and differently, a female and multiple males (again with sperm competition). These are the two types of ‘group-spawning’ in both hermaphroditic and gonochore species. Such information appears in many publications and is not exceptional or rare.

The reviewer is right that Table 1 did not capture all exceptions. However, we think Table 1 is useful since, to the best of our knowledge, such a table does not exist anywhere and therefore we would like to keep it. However, we have substituted the male and female symbols in the columns about mating system and spawning behavior with the text that was in the footnotes and have added different possibilities when needed. In this way doubts are eliminated and, in addition, spares readers going back and forth from the table to the footnotes.

Reviewer #3

I found the revised version of this paper to be substantially improved and am now convinced that this is an important study that provides a strong basis for future investigations of the controls of sexual system evolution in fishes and other organisms. My only substantial comment is that the authors conclude that hermaphroditism does not evolve directly from gonochorism. I agree that the low transition rates support this conclusion but the authors could also directly test this within BayesTraits by comparing models where the rates are set to 0 or not allowed to reversible models. I think this would be desirable given the importance of the result to their paper. On a minor note there is a formatting error in figure 1 in the first row where the symbols for the predicted mating systems are on the same line instead of on separate lines.

We thank the reviewer for considering our study important and a strong basis for future studies. Using Monte Carlo Markov Chain with reversible Jump (MCMC RJ) models, we found that the transition rate between gonochorism and simultaneous hermaphroditism are estimated to be 0 or extremely low; conversely transitions from gonochorism to sequential hermaphroditism and from sequential hermaphroditism to simultaneous hermaphroditism are substantially higher (Figure 4, Table 2 in our ms). We thus conclude that direct evolutionary changes between gonochorism and simultaneous hermaphroditism are very unlikely to occur and instead the intermediate step of sequential hermaphroditism is required.

To further support this conclusion, Reviewer 3 suggests that we compare our results using RJ models with those generated by models without RJ and models in which the direct transitions between gonochorism and simultaneous hermaphroditism are set to be 0. Such comparisons would indeed be useful if we were using approaches like maximum likelihood that return a single estimate per parameter. Indeed, the standard procedure in maximum likelihood would be to compare a model in which transitions are free to vary with a model in which they are fixed (to 0 in this case). However, MCMC does not aim at finding the (single) most likely value for each transition rate but rather to explore parameter space and identify all possible values that fit the data well - these values are returned in the posterior sample. As a result, models with values that do *not* fit the data well (i.e. here high transition rates between gonochorism and simultaneous hermaphroditism) are excluded and do not appear in the posterior sample.

We also note that the RJ procedure we used is designed specifically to test whether models in which some transition rates are set to 0, or equal to one another, better fit the data than models with higher values for those transition rates. Thus, the resulting RJ posterior distribution already incorporates the very test that the reviewer is asking for – if models with transition rates between gonochorism and simultaneous hermaphroditism estimated to be equal to 0 are a better fit, they are in the posterior samples; conversely, if models with high values for those transitions are not in the posterior sample is because they do not fit the data well.

Therefore, an MCMC analysis setting transition rates between gonochorism and simultaneous hermaphroditism to 0 will result in models very similar to those already in our RJ posterior samples but will miss out models with very low values for those transition rates. Conversely, an MCMC analysis without RJ will return very low values for transition rates between gonochorism and simultaneous hermaphroditism but fewer 0 values. As a result, a comparison between these two models without RJ is unsatisfactory because both contain some, but not all possible, values (i.e. 0 and low values) that fit the data well – instead both 0 and very low values are returned in our RJ posterior sample. Likewise, a comparison between either model without RJ and our RJ posterior is unsatisfactory because any difference in model fit is also determined by the values of the other parameters and model simplification which RJ can achieve.

Finally, we note that a key advantage of RJ is that its design allows us to also avoid overparameterization, a particularly useful feature in our study given that 2 states have relatively small sample sizes for the number of parameters to be estimated (i.e. protandry: N=36; simultaneous hermaphroditism: N=46; 6 parameters to be estimated, i.e. 3 transition rates in and 3 outs of each state). Indeed, a preliminary analysis of an early version of the dataset failed to converge without RJ.

In conclusion, the analyses proposed by the reviewer will not satisfactorily address the point the reviewer raised. Instead, the results we provide and the approach we used do already provide the answer to the issue raised. Should however the Editor and Reviewer feel that these analyses are needed, we will require a substantial extension. The RJ analysis we present currently takes a few weeks to run due to the large sample of species (N=4598 at 4 states) and model complexity. In our experience, models without RJ take longer to run because they do not benefit from the model

simplification that RJ can achieve. Furthermore, we would need to estimate model fit across the posterior using Bays Factors. This requires estimation of harmonic means for each proposed model using stepping stone sampler; this procedure typically increases the duration of the analyses by x3 or x4 once appropriate settings have been achieved. We thus estimate that running the suggested analyses will take several months, assuming models without RJ converge.

Re. Figure 1, the reviewer must mean Table 1. Thank you for pointing this out. Table 1 has been redone and now it only contains text, as explained in the answer to reviewer 1.